# Hic-5 drives epithelial mechanotransduction promoting a feed-forward cycle of bronchoconstriction

Chimwemwe Mwase[1], Wenjiang Deng[1], Hyo Jin Kim [1], Jennifer A. Mitchel [1], Thien-Khoi Phung [1], Michael J. O'Sullivan [1], Joel A. Mathews [2], Jeffrey Crosby[2], Christopher E. Turner[3], Adam L. Haber [1] & Jin-Ah Park [1]✉

Mechanical forces are essential for organ function, but excessive or dysregulated forces can promote pathologic conditions. In asthma, bronchoconstriction narrows the airway, compressing the airway epithelium and activating mechanotransduction, yet key regulators of mechanotransduction remain unclear. Here we show that Hic-5, a focal adhesion adaptor, is a key regulator of epithelial mechanotransduction. In human airway epithelial cells at air–liquid interface exposed to mechanical compression that mimics bronchoconstriction, we find that compression induces Hic-5 expression in airway basal cells. We further validated these in vitro findings by reanalyzing single-cell RNA-seq data from patients with asthma undergoing bronchoconstriction after allergen challenge, which revealed increased Hic-5 expression in airway basal cells. Hic-5 knockdown in human airway epithelial cells markedly attenuates mechanoresponses to compression, including stress fiber formation, differential gene expression, and increased secretion of endothelin-1 (ET-1). Through secretion of ET-1, a potent bronchoconstrictor, Hic-5 drives epithelial mechanotransduction and promotes a feed-forward cycle of bronchoconstriction, thereby highlighting dysregulated mechanical forces as active drivers of human disease.

The confluent layer of endothelial and epithelial cells that lines the surfaces and lumens of internal organs is often exposed to excessive mechanical forces. These mechanical forces are linked to the pathophysiology of diseases in the heart, pulmonary aorta, esophagus, and lung[1]. In the lung, distension during ventilator-induced lung injury and airway epithelial compression during asthma exacerbations have been recognized as contributors to disease progression. Specifically, bronchoconstriction during asthma exacerbations activates mechanotransduction in airway epithelial cells, leading to alterations in the transcriptome and secretome, as well as the induction of cell death[2–5]. These mechanically induced responses play a critical role in the progression of asthma[6,7]. Worldwide, asthma exacerbations pose a significant clinical burden, contributing to hospitalizations, intensive care unit admissions, and approximately 1000 deaths per day[8]. While most studies have focused on airway inflammation driven by immune cell activation, the contribution of bronchoconstriction is often underappreciated. However, as described above, emerging evidence indicates that the excessive mechanical forces generated during bronchoconstriction not only damage epithelial cells but also amplify airway inflammation and remodeling[5,7]. This mechanically initiated process may contribute to a vicious cycle that drives sustained and delayed asthmatic responses, the condition frequently experienced by patients with severe asthma[9]. Therefore, elucidating the precise mechanisms by which these mechanical forces are transduced into

[1]Department of Environmental Health, Harvard T.H. Chan School of Public Health, Boston, MA, USA. [2]Ionis Pharmaceuticals, Carlsbad, CA, USA. [3]Department of Cell and Developmental Biology, SUNY Upstate Medical University, Syracuse, NY, USA. ✉e-mail: jpark@hsph.harvard.edu

pathological responses in airway epithelial cells is essential for identifying new therapeutic targets and advancing the treatment of asthma. Moreover, uncovering how mechanical forces trigger inflammation and tissue remodeling likely has broader implications for disease in other organs where controlled mechanical forces are needed to maintain normal physiological functions.

To fill this gap, and as detailed in our experimental results, we investigated the role of Hic-5 (encoded by *TGFB1I1*), a focal adhesion adapter protein[10,11], in airway epithelial cells. Using RNA sequencing (RNA-seq) analysis of well-differentiated primary human bronchial epithelial (HBE) cells exposed to mechanical compression, mimicking epithelial deformation during bronchoconstriction, we identified *TGFB1I1* as a key mechanosensitive gene in airway epithelial cells. Hic-5 protein levels were also induced by mechanical compression, and the induction was dependent on the TGF-β receptor and ERK signaling. Single-cell RNA-seq (scRNA-seq) analysis of cells isolated from asthma patients revealed elevated *TGFB1I1* expression in basal airway epithelial cells and confirmed its link to bronchoconstriction. Hic-5 knockdown (KD) in HBE cells attenuated mechanoresponses, reducing differentially expressed genes (DEGs) by over 70%, including *EDN1*. This attenuation of *EDN1* expression led to decreased secretion of its encoded protein, endothelin-1 (ET-1), a potent bronchoconstrictor. Furthermore, Hic-5 KD protected against stress fiber formation, highlighting its role in the regulation of cellular force generation. These findings identify Hic-5 as a key regulator of airway epithelial

mechanotransduction, linking mechanical stress to gene regulation and airway remodeling in asthma.

## Results
### TGFB1I1 expression is induced by mechanical compression and TGF-β1 in airway epithelial cells in vitro and during asthma exacerbations in humans

While the role of Hic-5 has been studied in the context of epithelial-to-mesenchymal transition (EMT) and cancer metastasis, its role in mechanically dysregulated lung conditions, particularly those associated with progressive pathological conditions, has not been previously explored, highlighting our findings[12–14]. In our RNA-seq analysis in HBE cells, we identified that mechanical compression significantly induced *TGFB1I1*, the gene encoding Hic-5 protein. In HBE cells from all six donors without underlying airway disease, *TGFB1I1* was significantly upregulated by mechanical compression at both 3 and 24 hours post-compression (Fig. 1A). To validate our RNA-seq data, we performed a time-course experiment by measuring *TGFB1I1* mRNA expression by qPCR at 3, 8, and 24 h post-compression (Supplementary Fig. 1). We observed a transient increase in *TGFB1I1* mRNA expression, which was maximally induced at 3 h post-compression, with its significant induction of 4.2-fold (Fig. 1B) compared to the time-matched control. As *TGFB1I1* was originally identified as a TGF-β1-inducible gene, and its induction by TGF-β1 has been observed in multiple cell types[15], we tested whether it is similarly induced by TGF-β1 in HBE cells, given that

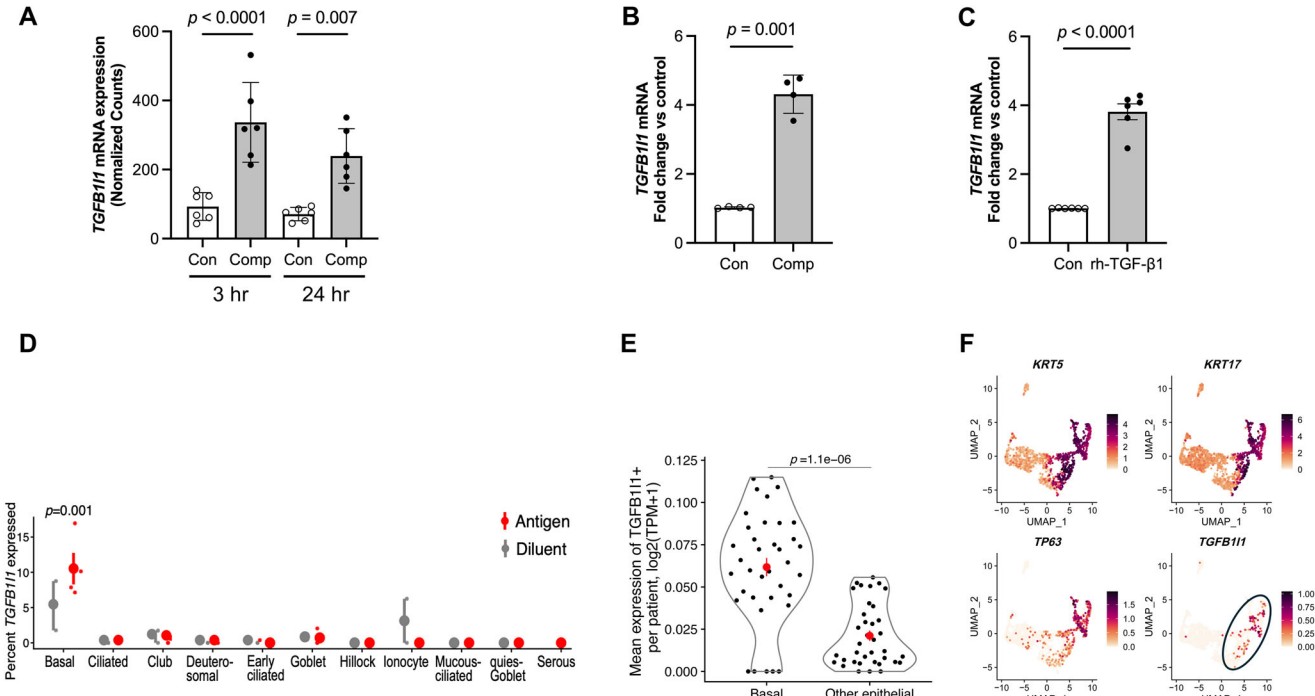

**Fig. 1 | The expression of *TGFB1I1*, encoding Hic-5, is induced by mechanical compression, rhTGF-β1, and bronchoconstriction in human airway basal cells.**
**A** Compression-induced *TGFB1I1* mRNA expression from scRNA-seq data. Expression plots show normalized *TGFB1I1* mRNA levels at 3 and 24 h following mechanical compression (30 cm H$_2$O). *TGFB1I1* mRNA expression was significantly increased by mechanical compression (mean ± SEM, n = 6 HBE cell donors). *P* values: one-way ANOVA with Holm–Šídák post hoc correction.
**B** *TGFB1I1* mRNA expression in HBE cells was measured at 24 h post-compression. *TGFB1I1* mRNA expression was significantly increased by mechanical compression (mean ± SEM, n = 4 HBE cell donors) *P* value: two-way ANOVA. **C** *TGFB1I1* mRNA expression in HBE cells was determined following exposure to rhTGF-β1 (10 ng/ml) for 24 h. Stimulation with rhTGF-β1 significantly increased *TGFB1I1* expression (mean ± SEM, n = 6 HBE cell donors). *P* value: two-way ANOVA. In **A**–**C**, each symbol represents an individual donor: open circles for control and closed circles for

compression or rhTGF-β1. **D** Reanalysis of published scRNA-seq data from human bronchial biopsies[17]. The plot presents the percentage of cells (y-axis) in each epithelial cell type (x-axis) expressing *TGFB1I1* for individual patients (small dots) after challenge with a diluent control (gray circles) or an allergic antigen (red circles). *P* value: MAST likelihood-ratio test, two-sided[56]. (n = 4 patients).
**E** Reanalysis of published scRNA-seq data from the Human Lung Cell Atlas[18] comparing mean expression of *TGFB1I1* transcript (y-axis) between airway basal and non-basal (other lung epithelial) cells (x-axis) for each individual donor (small dots). *P* value: Wilcoxon rank-sum test. (n = 37 donors) In **D**, **E**, large dots represent the mean of individual human donors, mean ± SEM. **F** Uniform manifold approximation and projection (UMAP) embedding of published scRNA-seq profiles of human airway epithelial cells after 14 days of air-liquid interface (ALI) differentiation, downloaded from GEO accession (GSM2772992)[19], colored by expression of key marker genes.

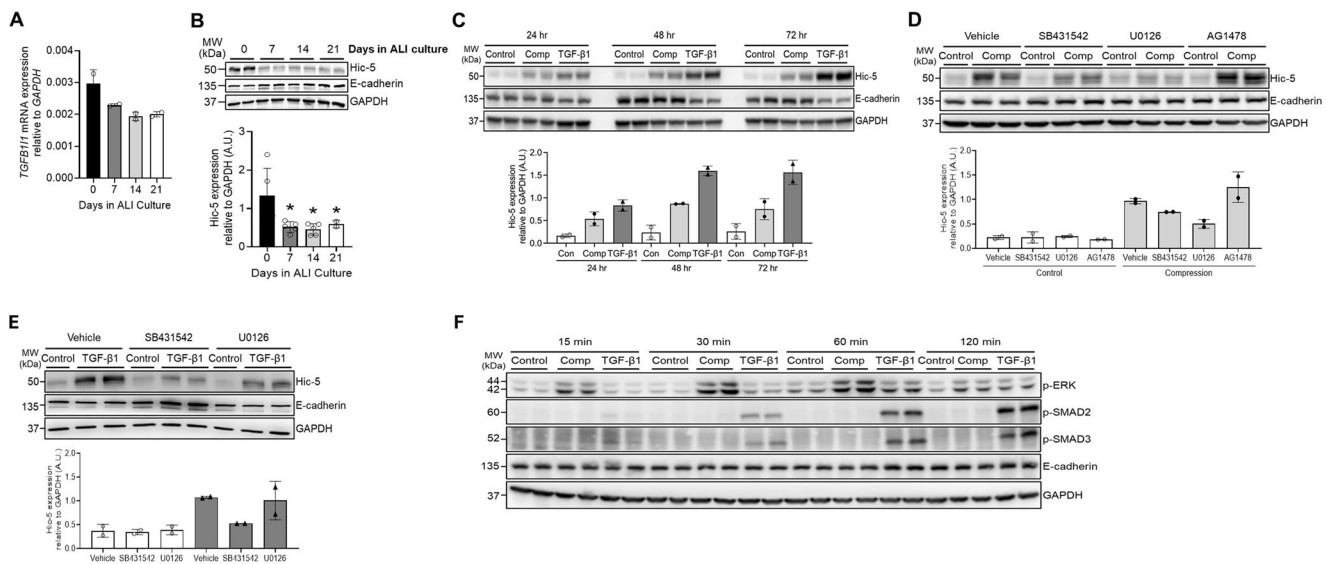

**Fig. 2 | The induction of Hic-5 depends on the activation of TGF-β receptor and ERK. A** *TGFB1I1* mRNA expression progressively decreased as basal cells differentiated over the course of 21 days in ALI culture (mean ± SD, *n* = 2 transwells from a single HBE cell donor). **B** Representative western blots present that Hic-5 protein was prominently detectable in confluent basal cell layers on ALI day 0, and its expression decreased as the basal cells differentiated. Densitometry analysis was performed, and the fold change relative to GAPDH was calculated in arbitrary units (A.U.) (mean ± SD, *n* = 5 HBE cell donors). One-way ANOVA with Holm–Šídák post hoc correction (vs. ALI day 0): *\*p* = 0.0426 (ALI day 7), *\*p* = 0.0328 (ALI day 14), and *\*p* = 0.0328 (ALI day 21). In **A**, **B**, each open circle represents an individual donor. **C** In well-differentiated HBE cells, Hic-5 protein increased in response to both mechanical compression and rhTGF-β1 treatment. Compression-induced Hic-5 protein expression remained constant for up to 72 h post-compression, whereas rhTGF-β1 stimulation progressively increased Hic-5 protein levels over 72 h. Densitometry analysis was performed, and the fold change relative to GAPDH was calculated in arbitrary units (A.U.) (mean ± SD, *n* = 2 HBE cell donors). **D** Representative western blots demonstrate that the compression-induced Hic-5 expression was attenuated by SB431542 (TGF-β receptor inhibitor) and U0126 (ERK

inhibitor) but remained unaffected by AG1478 (EGFR inhibitor). E-cadherin and GAPDH were detected as loading controls. Densitometry analysis was performed, and the fold change relative to GAPDH was calculated in arbitrary units (A.U.) (mean ± SD, *n* = 2 HBE cell donors). **E** Representative western blots demonstrate that the TGF-β1-induced Hic-5 expression was attenuated by SB431542 and U0126. E-cadherin and GAPDH were detected as loading controls. Densitometry analysis was performed, and the fold change relative to GAPDH was calculated in arbitrary units (A.U.) (mean ± SD, *n* = 2 HBE cell donors). In **C**−**E**, each symbol represents an individual donor corresponding to the conditions indicated by the bars.
**F** Representative western blots present the time course of phosphorylation of SMAD2, SMAD3, and ERK (p-SMAD2, p-SMAD3, and p-ERK) in HBE cells following stimulation with mechanical compression or TGF-β1. Compression prominently increased p-ERK levels as early as 15 min and sustained up to 60 min post-stimulation, but did not induce either p-SMAD2 or p-SMAD3. TGF-β1 substantially increased p-SMAD2 and p-SMAD3 within 30 min and sustained up to 120 min, while modestly increased p-ERK levels detected at 60 minutes (*n* = 2 HBE cell donors). Uncropped versions of the blots are provided as a Source Data file.

TGF-β1 levels are elevated in patients with asthma. After incubation of HBE cells with recombinant human TGF-β1 (rhTGF-β1, 10 ng/ml) for 24 h, we detected a significant 3.8-fold induction of *TGFB1I1* mRNA expression (Fig. 1C).

To further establish a mechanistic link between *TGFB1I1* induction in our in vitro system and its clinical relevance during bronchoconstriction in asthma exacerbations, we employed a unique approach by annotating *TGFB1I1 expression using* scRNA-seq data from published human asthma exacerbation studies[16]. This analysis revealed a significant induction of *TGFB1I1* expression, predominantly in basal airway epithelial cells, following allergen challenge (Fig. 1D). Importantly, the extent of allergen-induced *TGFB1I1* expression was significantly greater in individuals with asthma who underwent bronchoconstriction than in allergic individuals without asthma (Supplementary Fig. 2). In addition, no significant difference in *TGFB1I1* expression was detected at baseline between allergic controls and individuals with allergic asthma. This suggests that elevated Hic-5 expression is not a constitutive feature of the asthmatic airway epithelium, but rather a response to allergen exposure or bronchoconstriction, and may represent a critical regulatory mechanism in uncontrolled asthma. We then validated this cell-type-specific expression of *TGFB1I1* in basal cells by analyzing two additional publicly available scRNA-seq datasets[17,18]. In the first scRNA-seq dataset[17], we assessed the regional distribution of *TGFB1I1* expression across the respiratory tract, including the nasal passages, airways, and lung parenchyma (Supplementary Fig. 3). Our analysis demonstrated the expression of *TGFB1I1*

in both the airway and lung parenchyma, but not in the nasal passages. Within the airway, *TGFB1I1* expression was significantly higher in basal cells compared to non-basal cell populations (Fig. 1E). In the second dataset[18], which is well-annotated scRNA-seq data from the Human Cell Atlas, *TGFB1I1* expression was enriched in cells annotated by basal cell-specific markers, such as *TP63*, *KRT5*, and *KRT17* (Fig. 1F). Both analyses confirmed predominant expression of *TGFB1I1* in basal cells in human airways.

### Hic-5 protein induction is mediated by the TGF-β receptor-ERK axis in response to mechanical compression and TGF-β1

Consistent with *TGFB1I1* being annotated in basal cells, our qPCR analysis also detected a substantial level of *TGFB1I1 mRNA expression* at the time the air-liquid interface (ALI) culture was established (on ALI day 0), which subsequently decreased over the course of differentiation in ALI culture (Fig. 2A). Unlike this reduction during differentiation, expression of the basal cell marker gene *TP63* remained unchanged, whereas expression of differentiation markers, *MUC5AC*, *FOXJ1*, and *DNAI2*, increased (Supplementary Fig. 4). Hic-5, the protein encoded by *TGFB1I1*, was detectable by western blot analysis in confluent basal cell layers but significantly decreased over the course of ALI culture (Fig. 2B). Similar to the induction of *TGFB1I1* expression (Fig. 1A, B), Hic-5 protein expression was induced by both mechanical compression and TGF-β1 (Fig. 2C). The induction of Hic-5 by mechanical compression was detected at the first 24 h post-compression and remained constant with no further induction up to

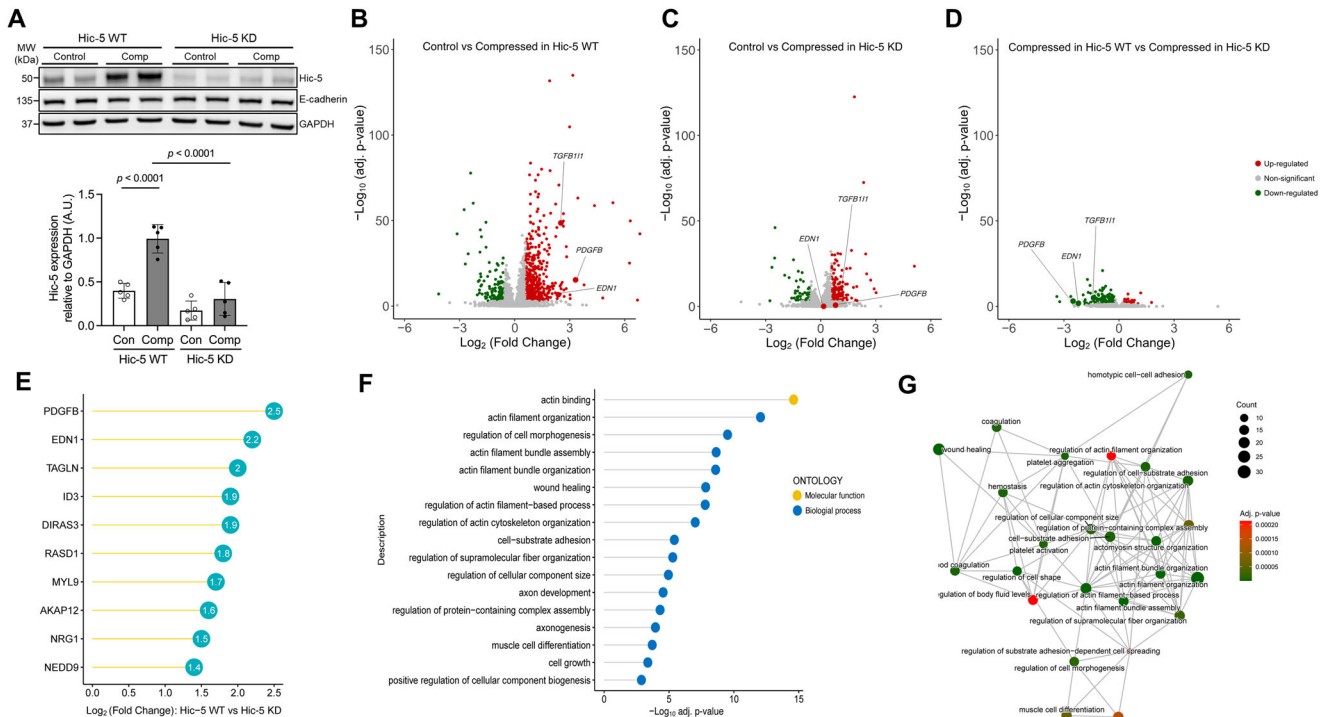

**Fig. 3 | Mechanical compression regulates multiple cellular responses in a Hic-5-dependent manner. A** Representative Western blots demonstrate the effective knockdown (KD) of Hic-5 in bronchial epithelial cells at baseline and in response to mechanical compression. Compression induced Hic-5 expression in Hic-5 wildtype (WT) cells, whereas this Hic-5 induction was abolished in Hic-5 KD cells. Densitometry analysis was performed, and the fold change relative to GAPDH was calculated in arbitrary units (A.U.) (mean ± SD, $n = 5$ HBE cell donors). $P$ values: one-way ANOVA with Holm–Šídák post hoc correction. Uncropped versions of the blots are provided as a Source Data file. **B**, **C** Volcano plots present differentially expressed genes (DEGs) between control and compressed conditions (DESeq2 test, two-sided). In Hic-5 WT cells, mechanical compression induced a significant number of

DEGs (**B**). In Hic-5 KD cells, the number of DEGs were significantly fewer compared to WT cells (**C**). $n = 3$ HBE cell donors. **D** The volcano plot presents DEGs between Hic-5 WT and Hic-5 KD cells under compression. In **B**–**D**, upregulated genes are marked by red, downregulated genes by green, and not significant genes by gray. **E** The top 10 DEGs were ranked by $Log_2$ (fold change) between Hic-5 WT vs Hic-5-KD under compression. **F** Gene Ontology (GO) analysis identified biological processes and molecular functions of upregulated genes in Hic-5 WT cells compared with Hic-5 KD under compression, as demonstrated in (**D**). **G** The enrichment network visualized functional modules of upregulated genes in Hic-5 WT cells compared with Hic-5 KD under compression, as demonstrated in (**D**).

72 h. In contrast, the induction of Hic-5 by TGF-β1, which was detected at 24 h continued to increase in a time-dependent manner, showing further elevation at 48 and 72 h (Fig. 2C). These differences in the kinetics of Hic-5 protein induction may be attributed to the source of TGF-β1, with autocrine production by mechanically compressed epithelial cells versus exogenous TGF-β1 treatment, which mimics its presence in the extracellular environment under chronic disease conditions.

To elucidate the signaling pathways regulating the induction of the Hic-5 protein, we tested inhibitors targeting key signaling molecules activated by mechanical compression, including the TGF-β receptor, ERK, and EGFR, which we previously identified[2,19–21]. To block each pathway in mechanically compressed cells, we pretreated the cells with pharmacological inhibitors: SB431542 for the TGF-β receptor, U0126 for ERK, and AG1478 for EGFR. Pretreatment with SB431542 and U0126 attenuated compression-induced Hic-5 protein expression, indicating that compression-induced Hic-5 expression was dependent on the ERK and TGF-β receptor pathways. In contrast, pretreatment with AG1478 had no effect, indicating that compression-induced Hic-5 expression was independent of the EGFR pathway (Fig. 2D). Similarly, pretreatment with SB431542 and U0126 attenuated TGF-β1-induced Hic-5 expression (Fig. 2E), confirming that the TGF-β receptor-ERK axis serves as a common mechanism for Hic-5 induction in response to either mechanical compression or TGF-β1. Given that the TGF-β receptor-ERK axis mediates Hic-5 activation under both conditions, we further examined the downstream TGF-β receptor pathways activated by compression and TGF-β1. We detected p-SMAD2 and p-SMAD3,

indicative of the canonical TGF-β receptor pathway, and p-ERK, indicative of the non-canonical TGF-β receptor pathway (Fig. 2F). As expected, TGF-β1 treatment induced both p-SMAD2 and p-SMAD3 within 30 minutes, followed by a moderate increase in p-ERK at later time points. In contrast, compression markedly induced p-ERK with no noticeable induction of either p-SMAD2 or p-SMAD3.

## Hic-5 regulates mechanoresponsive genes in HBE cells

Our data, as demonstrated in Fig. 2, indicated that Hic-5 induction depended on key signaling pathways regulating asthma-associated mediators[2,19–21]. Given these mediators are induced by mechanical compression in HBE cells, we hypothesized that Hic-5 plays a regulatory role in airway epithelial mechanoresponses. To test this, we used an antisense oligonucleotide (ASO)-mediated KD approach in well-differentiated HBE cells. Among the five ASO sets we tested in primary HBE cells cultured from multiple donors, we selected the ASO set demonstrating the highest KD efficiency of Hic-5, both at baseline and in response to mechanical compression (Supplementary Fig. 5). In HBE cells treated with a control (scrambled) ASO (denoted as Hic-5 WT), mechanical compression substantially induced Hic-5 protein levels (Fig. 3A). In contrast, in HBE cells treated with an ASO targeting Hic-5 (denoted as Hic-5 KD), where Hic-5 protein levels were markedly lower than in Hic-5 WT cells, the compression-induced increase in Hic-5 protein was minimal, resulting in levels even lower than those detected under control conditions in Hic-5 WT cells (Fig. 3A). Using this ASO-mediated KD approach combined with bulk RNA-seq analysis, we next investigated how Hic-5 deficiency modulates transcriptional

profiles in HBE cells at baseline and in response to mechanical compression. First, we compared Hic-5 WT and KD HBE cells under control conditions to confirm the specificity of ASO-mediated KD and to assess potential off-target effects. RNA-seq analysis revealed no other DEGs between Hic-5 WT and KD cells, except for the significantly downregulated *TGFB1I1*, which encodes the Hic-5 protein (Supplementary Fig. 6). This result confirms the selectivity of the ASO-mediated Hic-5 KD with minimal off-target effects. We then identified DEGs between control and compressed conditions at 3 hr post-compression in both Hic-5 WT (Fig. 3B) and Hic-5 KD (Fig. 3C) HBE cells. In WT HBE cells, 445 genes were upregulated, and 159 genes were downregulated (absolute log$_2$-fold change >0.6 and FDR <0.001, Fig. 3B). In contrast, Hic-5 KD cells displayed a markedly attenuated response to compression, with only 136 upregulated and 49 downregulated genes (Fig. 3C). *TGFB1I1* expression was significantly increased by 5.7-fold (FDR <0.001) in Hic-5 WT. While its expression appeared to be significantly induced by 1.2-fold in Hic-5 KD cells, this was likely due to a substantial reduction in *TGFB1I1* baseline levels, as shown in Fig. 3A. To compare the effect of Hic-5 KD on compression-mediated transcripts, we also illustrated the DEGs between compressed Hic-5 WT and compressed Hic-5 KD cells using a volcano plot (Fig. 3D) and heatmaps (Supplementary Fig. 7). The comparison showed that 193 compression-induced genes were downregulated in Hic-5 KD cells compared with Hic-5 WT cells under compression, suggesting attenuated induction of compression-responsive genes in the absence of Hic-5. These genes included *TGFB1I1* (encoding Hic-5), *PDGFB*, and *EDN1*, as annotated in Fig. 3D.

To identify candidate genes regulated by Hic-5, we ranked the downregulated genes by log$_2$ fold change values (Fig. 3E). The top-ranked genes included *PDGFB*, which had a log$_2$ fold change of −2.5 in Hic-5 KD cells, followed by *EDN1* with a log$_2$ fold change of −2.2. In Fig. 3B, C, we annotated *PDGFB* and *EDN1* to highlight their responses to mechanical compression in Hic-5 WT and Hic-5 KD HBE cells. In WT, *PDGFB* and *EDN1* showed significant increases of 9.9- and 5.3-fold, respectively (Fig. 3B). However, in Hic-5 KD cells, the expression of both genes showed no significant differences between control and compressed cells (Fig. 3C, D). To explore the biological pathways affected by Hic-5 KD, we performed gene ontology (GO) analysis using the downregulated genes (Fig. 3F). The pathways modulated by Hic-5 KD include actin binding, actin cytoskeleton organization, and regulation of cell morphogenesis, all of which are related to cellular differentiation, morphogenesis, and migration. To examine the overlap and potential interactions between these activated pathways, we generated a gene enrichment network plot (Fig. 3G), which enabled more detailed analysis of the pathway modules affected by Hic-5 KD. The first module involved actin filament and structure programs, including the genes *NEDD9*, *LCP1*, and *ACTN1*, suggesting that Hic-5 KD may disturb the normal actin filament functions[22]. The second module involved cell shape regulation, cell-substrate adhesion, and wound healing-related programs, including the genes *PDGFB*, *EDN1*, and *FERMT2*, implicating that Hic-5 may play a role in maintaining the structural integrity of the airway barrier[23]. Lastly, the third major module involved platelet activation, hemostasis, and coagulation programs, including the genes *PDGFB*, *EDN1*, and *F3*, indicating that Hic-5 is needed for hemostatic activity in the lung[24].

## Hic-5 deficiency protects against stress fiber formation and cell shape changes in response to mechanical compression

We have previously demonstrated that mechanical compression induces basal cell stress fiber formation that is associated with intercellular force generation during epithelial cell unjamming transition[25]. This stress fiber formation could be regulated by Hic-5 as indicated in our GO analysis (Fig. 3F). To validate the role of Hic-5 in stress fiber formation, we performed immunofluorescence (IF) staining for Hic-5 and F-actin in both Hic-5 WT and KD cells, with and without mechanical compression. In both conditions, Hic-5 localized predominantly to the

cytosol and did not accumulate in the nucleus following mechanical compression, although its overall expression level increased (Fig. 4A). Consistent with our previous findings, compression induced basal stress fiber formation accompanied by apical cell elongation in Hic-5 WT HBE cells (Fig. 4B and Supplementary Fig. 8). In contrast, these compression-induced changes were markedly attenuated in Hic-5 KD HBE cells, indicating that Hic-5 is essential for cytoskeletal remodeling triggered by mechanical compression.

## Hic-5 deficiency attenuates endothelin-1 secretion

We then further determined the role of Hic-5 in the regulation of *PDGFB* and *EDN1*, two top-ranked genes, as demonstrated in our RNA-seq analysis (Fig. 3E). We quantified *PDGFB* mRNA expression by qPCR in both WT and KD HBE cells. In Hic-5 WT HBE cells, mechanical compression markedly induced *PDGFB* mRNA expression by 9.6-fold compared to control (Fig. 5A). However, in Hic-5 KD HBE cells, compression-induced *PDGFB* expression was completely abolished. PDGF is known to be secreted from various cell types, including epithelial cells and detected at high concentrations in asthma[26,27]. Therefore, we also assessed the effect of Hic-5 deficiency on the secreted PDGF protein levels. However, secreted PDGF protein was undetectable by ELISA (R&D ELISA kit) under all conditions. We next validated *EDN1*, the second top-ranked gene highlighted in Fig. 3E, which encodes ET-1 protein. This validation was particularly intriguing, as we previously reported that mechanical compression induces ET-1 protein secretion, which in turn stimulates airway smooth muscle (ASM) contraction[28]. Consistent with our prior findings, in Hic-5 WT HBE cells, mechanical compression significantly induced *EDN1* mRNA expression by 9.8-fold (Fig. 5B) and increased ET-1 secretion by 74% (Fig. 5C) compared to control. However, in Hic-5 KD HBE cells, the compression-induced *EDN1* mRNA expression was significantly attenuated by 88% and ET-1 secretion was similarly decreased by 65% compared to the compressed condition in Hic-5 WT cells (Fig. 5B, C). Moreover, ET-1 secretion induced by TGF-β1 in Hic-5 WT was also significantly abolished in Hic-5 KD (Fig. 5D).

## Discussion

In this study, we identified Hic-5, encoded by *TGFB1I1*, as a key regulator of mechanotransduction in human airway epithelial cells. Using bulk RNA-seq in well-differentiated HBE cells and reanalyzing published scRNA-seq data from human bronchial biopsies collected after allergen challenge[16], we established Hic-5 as a link between our in vitro findings and bronchoconstriction-induced epithelial responses in patients with asthma. Using HBE cells exposed to mechanical compression mimicking airway epithelial cell deformation during bronchoconstriction, we demonstrated that Hic-5 was induced through pathways dependent on the TGF-β receptor and ERK. Functional analysis using Hic-5 KD combined with RNA-seq revealed a significant reduction in DEGs between control and compressed HBE cells, indicating that Hic-5 plays a key role in transcriptional regulation under mechanical stress. Our data indicate that Hic-5 mediates key compression-induced responses, including actin cytoskeleton organization and ET-1 secretion, highlighting its role as a key link between mechanical stress on the airways and ASM contraction, a critical pathological component of asthma.

Our findings have important therapeutic implications for asthma treatment. Hic-5, a member of the paxillin superfamily, functions as a focal adhesion scaffold that transduces mechanical signals from extracellular stimuli to intracellular signaling cascades[11,29,30]. Hic-5 has been recognized as a regulator of cellular mechanoresponses in mostly mesenchymal or cancerous cells[12–14,29,31–34]. While its superfamily member protein paxillin has been studied for its role in epithelial injury responses through focal adhesion remodeling[35], the role of Hic-5 in epithelial tissues remains poorly defined. This limited understanding may be due to its low expression in mature epithelial cells, with levels

**A**

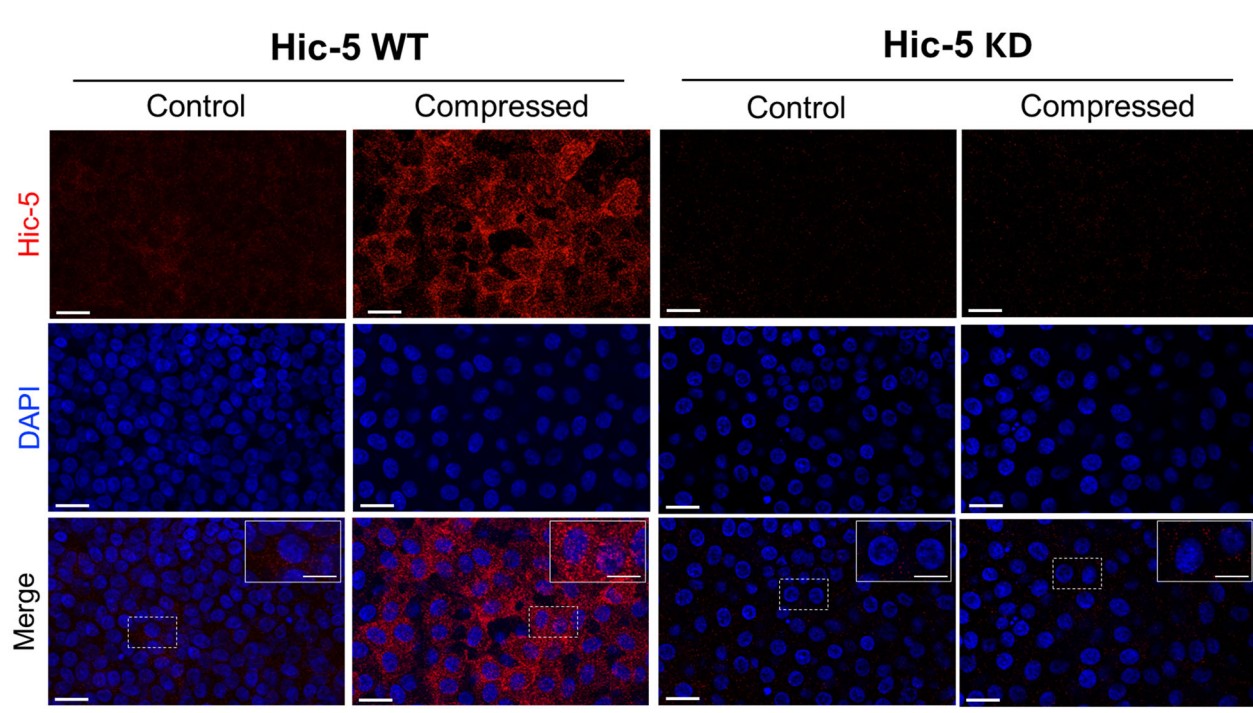

**B**

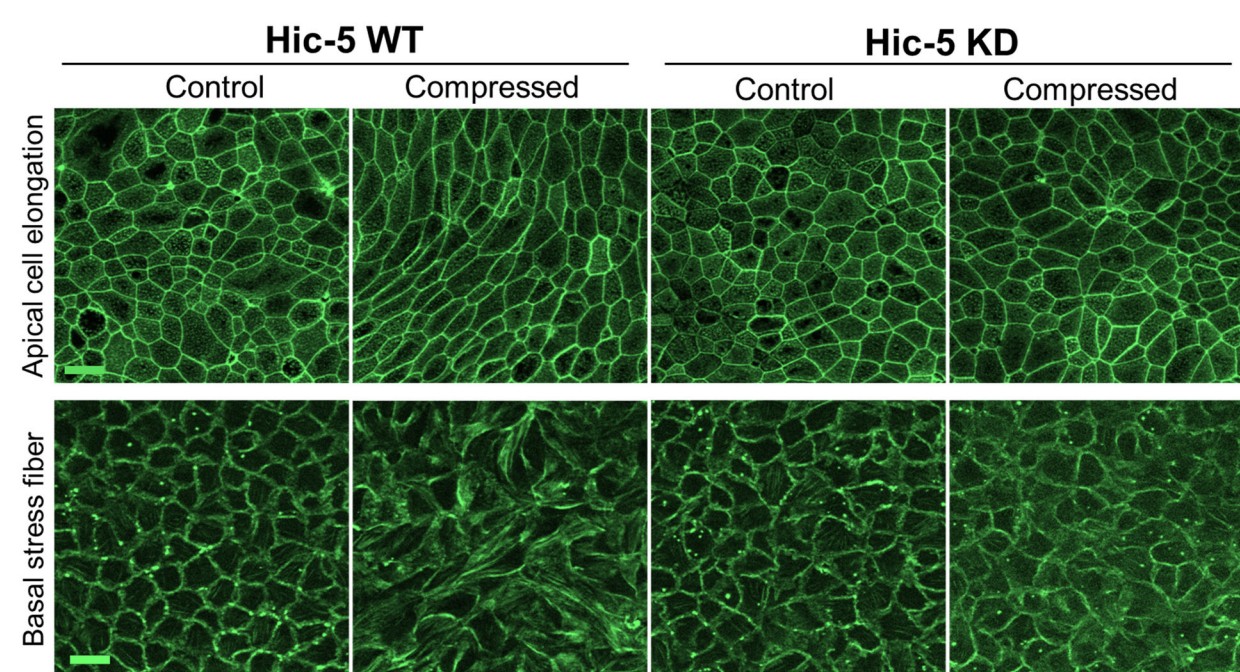

**Fig. 4 | Mechanical compression induces Hic-5, which then drives basal cell stress fiber formation and apical cell elongation. A, B** Representative immunofluorescence (IF) images of Hic-5 (red), nuclei (blue), and F-actin (green) in Hic-5 WT and Hic-5 KD HBE cells at 24 h post-compression. Mechanical compression induced the expression of Hic-5 protein, which localized to the cytosol in Hic-5 WT HBE cells but was abolished in Hic-5 KD cells (**A**). Dotted boxes indicate regions shown expanded twofold in the insets presented in the upper right corner. Scale bars = 20 μm (10 μm in insets). Mechanical compression induced apical cell elongation and basal cell stress fiber formation in Hic-5 WT HBE cells but not in Hic-5 KD HBE cells (**B**). Scale bar = 20 μm, $n = 5$ HBE cell donors.

increasing during EMT[13,36]. Consequently, Hic-5 has been studied more extensively in the context of cancer, where the EMT is a prominent feature[36–38]. Our findings provide direct evidence that mechanically induced Hic-5 initiates a cascade of events leading to airway constriction, a key component of the asthmatic response.

Our RNA-seq and qPCR analyses confirmed that mechanical compression significantly induced *TGFB1I1* expression in well-differentiated primary HBE cells (Figs. 1, 2). Since TGF-β1 signaling regulates epithelial proliferation, differentiation, extracellular matrix production, and inflammation, all of which contribute to asthma

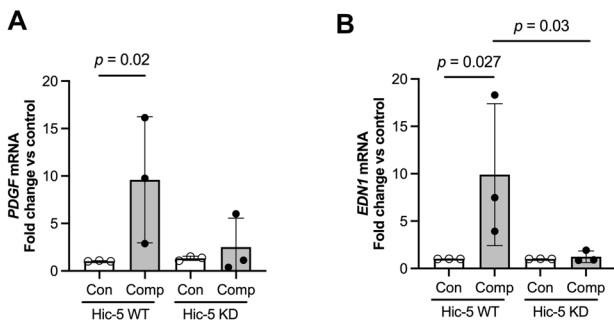

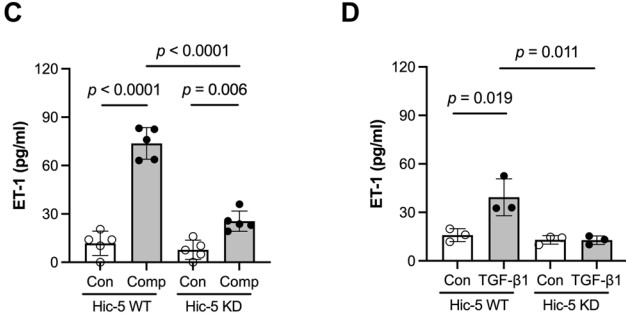

**Fig. 5 | Endothelin-1 expression and secretion depend on Hic-5. A, B** The validation of the top two ranked DE genes between Hic-5 WT and Hic-KD were quantified by qPCR. Compression-induced expression of *PDGF* (**A**) and *EDN1* (**B**) was observed in Hic-5 WT HBE cells but was abolished in Hic-5 KD cells (mean ± SD, *n* = 3 HBE cell donors). *P* values: two-sided Student's *t*-test. **C, D** Secretion of endothelin-1 (ET-1) by HBE cells into the basolateral conditioned media was determined by ELISA at 24 h post-compression (**C**) and post-TGF-β1 treatment (**D**). Hic-5 deficiency (Hic-5 KD) attenuated ET-1 secretion that was induced by both compression and TGF-β1 (mean ± SD, *n* = 5 HBE cell donors in (**C**) and *n* = 3 HBE cell donors in (**D**). *P* values: one-way ANOVA with Bonferroni post hoc correction. In **A–D**, each symbol represents an individual donor: open circles for control and closed circles for compression or rhTGF-β1 as indicated.

pathogenesis[39–43], Hic-5 may serve as a molecular bridge between mechanical and biochemical signaling in asthma. Elevated TGF-β1 levels in BALF, airway epithelium, and eosinophils correlate with disease severity[41,42,44], and TGF-β receptor activation contributes to airway remodeling and airway hyperresponsiveness (AHR) in both mice and humans[7,45,46]. Therefore, we hypothesize that Hic-5 acts as a key link between TGF-β receptor activation and cellular responses to both mechanical compression and TGF-β1 stimulation. scRNA-seq data revealed that *TGFB1I1* is enriched in basal cells, a progenitor population critical for airway repair and remodeling. *TGFB1I1* expression increased in basal cells from asthma patients experiencing bronchoconstriction following allergen exposure[16], linking Hic-5 to airway narrowing and remodeling. In addition, allergen-induced *TGFB1I1* expression was also detected in allergic controls, albeit to a lesser extent than in asthma patients, suggesting that additional allergy-mediated factors may contribute to the induction of Hic-5. No baseline difference was observed between allergic controls and individuals with allergic asthma. These findings suggest that elevated Hic-5 expression is not a constitutive feature of the asthmatic airway epithelium but rather part of an active response during asthma exacerbations or allergic conditions. Given its connection to asthma exacerbations, Hic-5 induction in airway epithelial cells may represent a critical regulatory mechanism in uncontrolled asthma. Our in vitro studies examined the independent effects of mechanical compression and TGF-β1. However, investigating their additive or synergistic interactions may provide further insight, as elevated TGF-β1 levels in the asthmatic airway environment, when combined with mechanical stress, could amplify Hic-5 expression and accelerate airway remodeling.

In ALI cultures, Hic-5 protein levels progressively decreased as basal stem cells differentiated, suggesting its potential roles in basal cell homeostasis, epithelial differentiation, and regeneration. Although well-differentiated HBE cells exhibited low baseline Hic-5 expression, it became inducible by mechanical compression and TGF-β1 in a manner dependent on TGF-β receptor and ERK signaling (Fig. 2). Interestingly, although our previous work demonstrated that the EGFR–ERK axis is required for YKL-40 and EGF ligand expression in mechanically compressed cells, Hic-5 induction is independent of EGFR activation[2,19]. Given this apparent disconnection of EGFR and ERK in Hic-5 regulation, we further explored whether ERK activation occurs through a TGF-βR-dependent pathway. TGF-β receptor signaling operates through both canonical (SMAD2/3) and non-canonical (ERK, JNK, MAPK, and ROCK) pathways[43]. In HBE cells, TGF-β1 stimulation predominantly activated the canonical SMAD2/3 pathway, while mechanical compression preferentially triggered non-canonical TGF-β-ERK signaling, despite both conditions converging on Hic-5 induction. Notably, TGF-β1-induced Hic-5 has been shown to be essential for sustained TGF-β1 production

through a feed-forward mechanism, in which persistently high TGF-β1 levels maintain the pathogenic myofibroblast phenotype observed in hypertrophic scar tissue[47].

Our GO and gene enrichment network analyses revealed that Hic-5 regulates key cellular responses to mechanical stress, including actin filament organization, cell shape, wound healing, and coagulation activity (Fig. 3). In a previous report using trabecular meshwork cells, TGF-β2 and ET-1 have been shown to induce Hic-5 expression, which in turn regulates actin cytoskeletal organization[12]. In our study, we observed that Hic-5 deficiency disrupted stress fiber formation, supporting its role in the dynamic regulation of cytoskeletal organization and cell shape changes. Stress fiber formation governs cellular responses to mechanical stress by increasing stiffness and promoting cell shape elongation, thereby increasing epithelial susceptibility to mechanical injury[35,48,49]. In A549 cells, transmural pressure leads to the activation of NFκB in a manner dependent on actin stress fiber formation, indicating a potential role of the actin cytoskeleton in mechanotransduction-mediated inflammatory signaling[50]. This may underlie the mechanism of epithelial cell death and subsequent inflammation observed in models of bronchoconstriction[5]. The association between Hic-5 and cell shape elongation is also reported in cancer cells[33], suggesting a broader role for Hic-5 in regulating cellular responses.

Our findings indicate that Hic-5 is essential for cytoskeletal remodeling in airway epithelial cells when exposed to mechanical stress. This cytoskeletal regulation, in turn, mediates mechanotransduction pathways that translate mechanical cues into transcriptional responses, including *EDN1* expression. In the pseudostratified human airway epithelium, mechanical compression mimicking bronchoconstriction increases ET-1 secretion independently of airway smooth muscle, suggesting that epithelial mechanotransduction alone is sufficient to drive ET-1 secretion. Hic-5 KD markedly suppresses this induction, supporting a role for Hic-5 as a critical regulator of epithelial ET-1 secretion under mechanical stress. These findings raise the possibility that a Hic-5–ET-1 axis contributes to a feed-forward cycle of bronchoconstriction in severe asthma and provide a rationale to explore epithelial-specific Hic-5 inhibition, for example, through inhaled ASO therapy. While our findings establish that Hic-5 is required for compression-induced transcriptional regulation, it remains uncertain whether Hic-5 overexpression alone can drive these changes in HBE cells. Although Hic-5 has been reported to act as a co-activator of transcription factors such as the androgen receptor, our findings show that it does not accumulate in the nucleus of airway epithelial cells. This suggests that Hic-5 does not directly activate transcription via nuclear translocation during compression-induced mechanotransduction. Instead, it likely contributes indirectly

by regulating cytoskeletal tension and actin dynamics, which can activate downstream mechanosensitive pathways such as YAP/TAZ or MRTF–SRF signaling[51,52]. Both pathways are well established to respond to actin stress fiber assembly and cytoskeletal tension, raising the possibility that Hic-5 functions as an upstream mediator of these mechanically regulated transcriptional processes in the airway epithelium. Prior studies in vimentin-rich fibroblasts[53] further suggest that Hic-5 functions mainly as a focal adhesion adapter, supporting the possibility that Hic-5 mediates mechanotransduction but is not sufficient on its own. Future studies will be needed to determine whether Hic-5 overexpression is sufficient and to identify adaptor(s) that may substitute for vimentin in vimentin-deficient well-differentiated airway epithelium.

Our study identifies Hic-5 as a key regulator of mechanotransduction in airway epithelial cells, linking bronchoconstriction-induced mechanical stress to asthma pathogenesis. Through its roles in cytoskeleton organization and transcriptional regulation, including ET-1, Hic-5 links acute asthma exacerbations to sustained asthmatic responses and airway remodeling, the hallmark of difficult-to-treat asthma. Together, our findings provide critical mechanistic insights into how mechanical stress shapes epithelial responses and highlight Hic-5 as a potential therapeutic target for mitigating disease progression in asthma. Moreover, our data advance our understanding of how dysregulated mechanical forces drive disease progression, offering mechanistic insights into chronic conditions governed by mechanotransduction beyond the lung.

## Methods

### Reanalysis of published single-cell RNA-seq data
Previously published scRNA-seq data from human asthma challenge model[16], human lung cell atlas[17], and human in vitro cultured airway epithelial cells[18] were downloaded from the respective relevant online repository and visualized using standard computational approaches.

### In vitro exposure of well-differentiated primary HBE cells to mechanical compression or TGF-β1
As previously described[3,4,19,20,28], we seeded primary HBE cells on transwells and maintained in ALI culture until the cells were well-differentiated. To mimic bronchoconstriction, we exposed well-differentiated HBE cells to apical-to-basal pressure with a magnitude of 30 cm $H_2O$ for 3 h, as previously described[3,4,19,20,28]. Time-matched control cells received 0 cm $H_2O$ pressure. We then collected cells and conditioned media between 3 and 72 h. In experiments using recombinant human (rh) TGF-β1 (Cell Signaling Technologies, Danvers, MA), we spiked 10 ng/ml into the basolateral media of HBE cells in ALI cultures and collected the cells and media at the indicated time points. To determine the signaling pathways that regulate mechanical compression-induced Hic-5 protein expression, we used pharmacological inhibitors of the TGF-β receptor (SB431542; 10 μM, Tocris, Bristol, UK), MEK (U0126; 10 μM, Tocris), and EGFR (AG1478; 10 μM, Tocris). Each inhibitor was added to the basolateral media of HBE cells 1 h prior to exposure to mechanical compression or TGF-β1. As a control, 0.1% DMSO was used.

### Hic-5 knockdown using antisense oligonucleotides (ASOs)
To knock down Hic-5 expression in well-differentiated primary HBE cells, we used antisense oligonucleotides (ASOs) designed and provided by Ionis Pharmaceuticals, following the previously established ASO approach[54]. We initially screened five ASOs targeting the coding region of human *TGFB1I1* (denoted as Hic-5 ASO). We then selected the most effective ASO based on the reduction of Hic-5 expression determined by both RT-qPCR and western blot analysis (Supplementary Fig. 5). We used this selected ASO targeting Hic-5 for subsequent experiments. As a non-targeting control, we used a scrambled ASO (denoted as control ASO) provided by Ionis Pharmaceuticals. We

reconstituted lyophilized ASOs in PBS to 5 mM, aliquoted the stock ASO, and stored the aliquots at -80 °C until use. During ALI culture, we replenished basal medium containing 10 uM ASO (scrambled or Hic-5) every other day from ALI day 9 to day 19.

### RNA sequencing analysis
As we previously described[3,4,19,20,28], we isolated total RNA from HBE cells ($n = 3$ donors without a history of lung disease) using the RNeasy Mini Kit (Qiagen, Hilden, Germany). Bulk RNA sequencing was performed on a NovaSeq instrument (Illumina, San Diego, CA, USA) by the Bauer Core Facility at Harvard University.

We evaluated the quality of RNA-seq data using MultiQC (v1.9) and FastQC (v0.12.0), and filtered out sequencing reads with low-quality metrics. Transcript expression at the isoform-level was quantified using Salmon (v1.10.1), and gene-level expression values were then aggregated from the isoform-level results. DEGs between the control and compressed groups were identified using DESeq2 (v1.42.0), enabling the detection of both upregulated and downregulated genes between Hic-5-WT and Hic-5-KD samples. DEGs were defined as adjusted $p$ value <0.05 and an absolute log2 (fold change) >0.25. The DEG results were visualized using volcano plots and heatmaps. Additionally, DESeq2 was employed to assess interaction effects between genotypes (i.e., Hic-5 WT vs. Hic-5 KD) and experimental conditions (i.e., control vs. compression). GO and pathway enrichment analyses were performed on significant DEGs from each comparison using clusterProfiler (v4.8.2). To facilitate efficient identification of functional modules, gene enrichment networks were constructed with the enrichplot package (v1.26.5). All data analyses and visualizations were conducted using R (v4.3.2) in RStudio (v2023.09.1).

### RT-qPCR
We performed real-time RT-qPCR using the primers (Supplementary Table 2) as previously described[4,19]. After normalization to *GAPDH*, the fold changes were calculated by the comparative $2^{-\Delta\Delta Ct}$ method[55]. To compare mRNA expression relative to *GAPDH* over the course of ALI culture, we calculated $2^{-\Delta Ct}$.

### Western blot analysis
We detected proteins in cell lysates by western blot analysis as previously described[4]. We used the following primary antibodies according to the manufacturer's instructions: Hic-5 (#6114) purchased from BD Biosciences (Woburn, MA); E-cadherin (#3195), GAPDH (#5174S), p-ERK (#4370S), p-SMAD2 (#3108), and p-SMAD3 (#9520) purchased from Cell Signaling Technology (Danvers, MA). We detected GAPDH as a loading control. Blot images were captured using Syngene GeneSys software (v1.7.9; Syngene, Cambridge, UK), and protein intensity was analyzed using ImageJ software (v1.46; National Institutes of Health, Bethesda, MD, USA).

### Immunofluorescence staining
Using our previously described immunofluorescence staining protocol[25], we stained cells for F-actin using phalloidin conjugated to AlexaFluor 488 (A12379; Thermo Fisher Scientific, Waltham, MA), at 24 h after compression. Cells were counterstained with Hoescht (33342; Thermo Fisher Scientific) for nuclei. Samples were imaged at 20× magnification using an Axio Observer fluorescence microscope (Zeiss, Danvers, MA, USA) and processed using Zeiss ZEN software (blue edition, v3.3) and ImageJ software (v1.46; National Institutes of Health, Bethesda, MD, USA).

### ELISA
We measured the amount of ET-1 protein released into basolateral conditioned media using a human-specific ELISA kit (DET100; R&D Systems, Minneapolis, MN), according to the vendor's instructions.

## Statistical analysis

All the data, except for RNA-seq data, were analyzed using GraphPad Prism (San Diego, CA). Statistical significance was determined using a two-tailed Student's $t$-test in experiments comparing two groups, or a two-way ANOVA followed by Bonferroni or Holm-Šídák post hoc corrections in experiments comparing three or more groups. A $p < 0.05$ was considered statistically significant.

## Ethics statement

All studies were conducted in accordance with relevant ethical regulations. The study protocol was reviewed and approved by the Committee on Microbiological Safety at Harvard University (approval no. 21-274). We used primary HBE cells isolated at the Marsico Lung Institute/Cystic Fibrosis Research Center, the University of North Carolina (UNC), Chapel Hill. Donor lungs were obtained under a protocol approved by the UNC Institutional Review Board (approval no. 03-1396). Informed consent was obtained from the legally authorized representatives of all organ donors. In this work, we used primary HBE cells from seventeen donors with no history of smoking or lung disease (Supplementary Table 1).

## Reporting summary

Further information on research design is available in the Nature Portfolio Reporting Summary linked to this article.

## Data availability

RNA sequencing data generated in this study have been deposited in the NCBI Gene Expression Omnibus (GEO) under accession number GSE310981, in accordance with the journal's guidelines. Source data for all graphs are provided with this paper as a Source Data file. Data that comprise the graphs within this manuscript and other findings of this study are available from the corresponding author upon request. Source data are provided with this paper.

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

## Acknowledgements

The authors thank Drs. Phyllis Kanki, Jeffrey Drazen, and Kari Nadeau for their critical review; Dr. Scott Randell and the Marsico Lung Institute/Cystic Fibrosis Research Center, Tissue Procurement and Cell Culture Core (The University of North Carolina, Chapel Hill) for providing primary human bronchial epithelial cells; and Dr. Tiffany Zhang for assistance with RNA-seq deposition. This study was funded by NHLBI: R01HL148152 (J.-A.P.), 5P01HL152953(J.-A.P.), T32HL007118 (J.A.M. and T.-K.P.), NIEHS: P30ES000002 (J.-A.P.), NIGMS: R35GM131709 (C.E.T.), FujiFilm Corporation (no grant number, C.M.), and Francis Family Foundation (no grant number, J.A.M.).

## Author contributions

C.M., J.A.M., M.J.O., J.-A.P. designed experiments; C.M., H.J.K., J.A.M., M.J.O., performed experiments, W.D., T.K.P., and A.L.H. performed RNA-seq analysis, C.M., H.J.K., J.A.M., T.-K.P., M.J.O., and J.-A.P. analyzed data; C.M., W.D., H.J.K., J.A.M., T.-K.P., M.J.O., J.M., J.C., C.E.T., A.L.H., and J.-A.P. interpreted data; C.M., W.D., A.L.H., and J.-A.P. drafted the manuscript, C.M., H.J.K., J.A.M., T.-K.P., M.J.O., J.M., J.C., C.E.T., A.L.H., and J.-A.P. reviewed the manuscript.

## Competing interests

The authors declare no competing interests.
