## [Transparent Peer Review file · Nature Communications]

Hic-5 drives epithelial mechanotransduction promoting a feed-forward cycle of bronchoconstriction

Corresponding Author: Dr Jin-Ah Park

Version 0:

Reviewer comments:

Reviewer #1

(Remarks to the Author)

The manuscript, Hic-5 transduces mechanical force that drives a vicious cycle of bronchoconstriction by Mwase, C et al. identifies a new signalling axis that is likely to be therapeutically useful for asthma, by focusing on mechanical signalling. The role of mechanics in pathophysiology has gained much ground these last decades, however, many key regulators of mechanotransduction remain poorly defined. Here, Park and colleagues uncover a novel and robust mechanoresponsive role for the focal adhesion adaptor protein, Hic-5, in airway epithelial cells. Using well-differentiated human bronchial epithelial cells in ALI cultures under compression, and previously published bulk and single-cell RNA sequencing datasets from asthmatics that underwent an allergen challenge, the authors set out to identify molecular targets regulated by bronchoconstriction. They found that Hic-5 (encoded by TGFB1/1) expression, mainly in basal cells, was significantly increased after mechanical compression. Notably, when Hic-5 was knocked down, the expression of more than 70% of the differentially regulated genes was abolished, highlighting its previously unknown role as a master transcriptional regulator of airway epithelial cell response to compression. Moreover, they show that endothelin-1 (ET-1), a potent bronchoconstrictor, expression and secretion was significantly increased by compression in a Hic-5-dependent manner, suggesting a pernicious feedforward cycle that could sustain bronchoconstriction and airway narrowing in asthmatics.

The authors provide strong evidence for a novel mechanical role of Hic-5 in human airway epithelial cells and define a pathological feedforward cycle of asthmatic bronchoconstriction that promotes further constriction via Hic-5 and ET-1. The experiments are well thought out, the writing lucid, and the data compelling, and we recommend its publication upon addressing the following minor concerns:

1. The authors show that TGFB1/1 is expressed either after compression or rhTGFB1 treatment (Fig 1). Does the combined treatment of compression and TGFB1 result in an additive or synergistic effect on TGFB1/1 expression?
2. The authors show that TGFB1/1 expression is increased in asthmatics after an exacerbation and not allergic controls, and this was seen mainly in the basal cell population. Further, they show that Hic-5 expression decreases after ALI day 0 from healthy HBEs. Could the authors speculate as to why this might be the case? If Hic-5 expression was sustained after ALI day 0 would they expect an asthmatic-like monolayer? Do ALIs from asthmatics have increased Hic-5 expression?
3. Figure 2 demonstrates that mechanically-induced Hic-5 expression requires ERK and TGFB receptor signalling, and this expression is required for a robust transcriptional response to mechanical stress (Fig 3). But how does Hic-5 activate a transcriptional response? The study could benefit from immunofluorescence analysis of Hic-5 before and after compression. Several studies have demonstrated a translocation of Hic-5 at focal adhesions to the nucleus after stimuli, like H2O2 and mechanical stress (Knockdown of mechanosensitive adaptor Hic-5 ameliorates post-traumatic osteoarthritis in rats through repression of MMP-13 - doi.org/10.1038/s41598-023-34659-x ; Hic-5 Communicates between Focal Adhesions and the Nucleus through Oxidant-Sensitive Nuclear Export Signal - doi: 10.1091/mbc.02-06-0099). Is Hic-5 actively activating transcription by entering the nucleus or does it signal through other mechanically responsive TFs, like YAP/TAZ or MRTFs?
4. The authors show in Fig 3 that Hic-5 is necessary for a robust transcriptional response to compression. Hic-5 is a homolog of Paxillin and found at focal adhesions, therefore, is this due to lack of Hic-5 or because focal adhesions have been

disrupted and mechanotransduction cannot occur efficiently from the cell surface? Fig. 4 could benefit from immunostaining of some of the adhesion complexes that Hic-5 regulates. Additionally, the images in Fig. 4 are not quantified, which is something that they could do, based on their previous studies on jamming and unjamming. Also, is Hic-5 over-expression sufficient to drive a compression-like transcriptional response without compression?

5. In the title the authors say that Hic-5 drives a vicious cycle of bronchoconstriction, one way being the stimulated secretion of ET-1 (Fig 5) in response to compression. However, the authors do not show any direct data relating to bronchoconstriction and it is first referenced in line 235-237 (Ref 28). Would the authors be able to either highlight or expand on this data to further support their title? If not, we recommend editing the title to not overstate the findings. Also, in Fig 5 Hic-5 KD shows near ablation of EDN1 mRNA after compression, however there is still a significant elevation of ET-1 protein after compression when comparing control and Hic-5 KD. Can the authors propose why there might still be residual ET-1 detected in Hic-5 KD after compression? Finally, the finding that ET-1 is much more tightly regulated by Hic-5 under compression without any smooth muscle is a finding that needs further highlighting in the discussion, as it could be the basis for therapeutic intervention.

Reviewer #2

(Remarks to the Author)

Reviewer #3

(Remarks to the Author)

Reviewer #4

(Remarks to the Author)

In this manuscript (NCOMMS-25-35454-T), Mwase and colleagues demonstrate that compression of airway epithelial cell cultures at an air-liquid interface induces the expression of Hic-5, a member of the paxillin family with multiple cellular functions. Notably, the protein is known to act as a scaffold at focal adhesions and may also have a role in transcriptional regulation via effects on nuclear receptors and other signaling pathways. It has been shown to have well-documented effects in multiple disease-relevant processes. Bronchoconstriction was felt to be a downstream effect of inflammation in asthma, but more recent data suggest that it can contribute to asthma pathophysiology by increasing inflammation and remodeling. The authors demonstrate expression of the gene for Hic-5 in human asthmatic airways and upregulation with allergen exposure (presumably following bronchoconstriction). Inhibition of the TGF β receptor and ERK mitigated the upregulation. They then demonstrate transcriptional changes following compression of epithelial cell cultures with knockdown of Hic-5. Prominent among the changes are reductions in PDGF β and endothelin-1 pathways. There are also associated morphologic changes in the cultures consistent with reduced mechanotransduction. The authors conclude that Hic-5 is a key regulator of mechanotransduction in airway epithelial cells and this links bronchoconstriction-induced mechanical stress to asthma pathogenesis. The data are clearly presented, and the manuscript is well written. The overall demonstration that Hic-5 is responsive to airway epithelial cell compression *in vitro* is convincing. The data from human studies does provide evidence that Hic-5 is expressed in basal cells and upregulated following an asthma-relevant stimulus, but it cannot be concluded that this is due to bronchoconstriction. The Hic-5 knockdown is impressive in the scope of effects, but the details of the methodology are limited. It is also not clear how well compression of cultures mimics the situation with bronchoconstriction *in vivo*. Finally, there is very little mechanistic data on how Hic-5 upregulates endothelin-1/PDGF β expression. Overall, the authors have identified an interesting mechanism driving Hic-5 expression that may link bronchoconstriction to airway remodeling, but the data are not fully supportive of their conclusions.

Specific points:

- 1) For the human correlation, the authors state that TGF β 111 levels were increased post-allergen in asthmatics but not in the control subjects. Are these data available? Were the differences significant?
- 2) Allergen challenge obviously induces multiple processes, including bronchoconstriction. How can you definitely say it was the bronchoconstriction in human subjects that induced the TGF β 111?
- 3) More details on the method of cellular compression (force used, % compressed etc) would be helpful. How well does this mimic the forces in the airway of an asthmatic that is constricted?
- 4) What are the antisense oligonucleotides utilized? How were they introduced into the cells and how specific are they to TGF β 111? Could there be off-target effects?

- 5) It would be helpful to quantify the band intensity in the western blots shown in Figures 2 and 3.
- 6) Is Figure 1A necessary, given the more quantitative data in 1B? Why not show the time course?
- 7) Are EDN1 and PDGFBeta upregulated in the human data, similar to TGFB111?
- 8) Can you link any of the known activities of Hic-5 to the signaling changes noted?
- 9) It is unclear to me why compressed WT cells don't have more dramatic DEG compared to compressed Hic-5 KO cells (Fig 3D), given the dramatic differences in the profiles when compared to control cells.

Version 1:

Reviewer comments:

Reviewer #1

(Remarks to the Author)

We have re-reviewed the Park paper and found that the authors addressed every comment that we raised satisfactorily. It is a beautiful paper and we recommend its publication.

Reviewer #2

(Remarks to the Author)

Reviewer #3

(Remarks to the Author)

Reviewer #4

(Remarks to the Author)

The revised manuscript and response to review have addressed almost all of my comments. One issue needs to be further clarified:

In the new extended data Figure 2 it appears that the AC subjects also have significant upregulation in TGFB111 in the allergen sample (p value listed is 0.03). The values look higher in the AA but its unclear if there is a difference between AA and AC. The legend implies that there is no change in AC which is not correct based on a p value of 0.03. This needs to be discussed in more detail. Similarly, I don't think the authors are acknowledging that there may be other factors with an allergen challenge that effect Hic-5. I think the discussion needs to acknowledge that the human allergen challenge data is merely suggestive and other factors induced by allergen may increase Hic-5, especially in light of the data now shown in the Extended figure 2.

Major Revision Decision on Nature Communications manuscript NCOMMS-25-35454-T

We thank the reviewers for their constructive comments and thoughtful suggestions. The reviewers' feedback was immensely helpful, and we are deeply appreciative. We believe that incorporating the reviewers' comments into the manuscript through our responses has substantially improved the clarity of the data presentation and the significance of the study. To address each critique in detail, we have prepared the rebuttal with the reviewers' comments reproduced verbatim and our point-by-point responses in blue. We have also carefully revised the manuscript to improve clarity and coherence. In the revised manuscript, **we highlighted substantive changes in red**. We cross-referenced them with line numbers in the rebuttal to guide the reviewers.

REVIEWER COMMENTS

Reviewer #1 (Remarks to the Author):

General comment

...

The authors provide strong evidence for a novel mechanical role of Hic-5 in human airway epithelial cells and define a pathological feedforward cycle of asthmatic bronchoconstriction that promotes further constriction via Hic-5 and ET-1. The experiments are well thought out, the writing lucid, and the data compelling, and we recommend its publication upon addressing the following minor concerns:

Response: We are grateful to the reviewer for taking the time and effort to review our manuscript. We are very encouraged by reviewer's positive comments and constructive suggestions. To address concerns raised by reviewer 1, we've prepared the detailed responses below and accordingly revised the manuscript. We value the reviewer's insights on various points, which have substantially helped improve the clarity of the data and overall quality of our revised manuscript.

Specific comments

Comment 1. *The authors show that TGFB1/1 is expressed either after compression or rhTGFb1 treatment (Fig 1). Does the combined treatment of compression and TGFb1 result in an additive or synergistic effect on TGFB1/1 expression?*

Response 1:

Thank you for this clinically relevant question. Because patients with asthma exhibit elevated TGF- β 1 levels in the lung microenvironment, the combined effect of mechanical compression and TGF- β 1 likely reflects the *in vivo* conditions. Although we did not include these experiments in the submitted manuscript, this is part of our ongoing studies investigating the underlying mechanisms of Hic-5 regulation. As shown in **Fig. R1**, co-stimulation with mechanical compression (30 cm H₂O) and rhTGF- β 1 (1 ng/ml) increased Hic-5 protein levels beyond either stimulus alone.

Figure R1. Co-stimulation with mechanical compression (30 cm H₂O) and rhTGF- β 1 (1 ng/ml) increased Hic-5 protein levels beyond either stimulus alone.

In Fig R1, we tested TGF- β 1 at 1ng/ml, an order of magnitude lower than the 10ng/ml used in the submitted manuscript to determine the combined effect with compression. These results demonstrate that combined stimulation enhanced Hic-5 protein expression, suggesting an additive or synergistic effect by co-stimulation. As **Fig. R1** is preliminary observations in HBE cells from a single donor, it is not sufficient to include in the current manuscript, but we have revised the text to incorporate this aspect in the revised manuscript (Lines 305 – 308).

Comment 2. *The authors show that TGFB1/1 expression is increased in asthmatics after an exacerbation and not allergic controls, and this was seen mainly in the basal cell population. Further, they show that Hic-5 expression decreases after ALI day 0 from healthy HBEs.*

2A. *Could the authors speculate as to why this might be the case?*

2B. *If Hic-5 expression was sustained after ALI day 0 would they expect an asthmatic-like monolayer?*

2C. *Do ALIs from asthmatics have increased Hic-5 expression?*

Response 2:

Thank you for insightful questions. We address each point (2A, 2B, and 2C) below:

2A. Hic-5 is well known for its role in promoting epithelial–mesenchymal transition (EMT). A recent study showed that Hic-5 expression is facilitated by a permissive chromatin environment, contributing to a cancer stem cell–like phenotype¹. During airway basal cell differentiation in ALI culture, cells undergo mesenchymal–epithelial transition (MET), the reverse of EMT. Accordingly, as basal stem cells differentiate into a well-structured epithelial layer, Hic-5 expression decreases, consistent with its association with EMT and cancer-promoting functions. Thus, we speculate that the reduction of Hic-5 expression during ALI culture reflects its role in regulating epithelial differentiation, as briefly noted in the Discussion (Lines 310 – 314).

2B. Given the role of Hic-5 in promoting EMT in epithelial cells, we anticipate that sustained Hic-5 expression beyond ALI day 0 could maintain a less differentiated, EMT-skewed epithelial state. Although the contribution of EMT to asthmatic airway epithelium remains somewhat controversial, such an EMT-skewed state could impair barrier function and favor basal cell–dominant, remodeling-prone features resembling an “asthmatic-like” monolayer. Therefore, sustained Hic-5 expression beyond ALI day 0 may maintain the airway epithelium in an asthma-like state.

2C. Our preliminary data suggest that the progressive reduction of Hic-5 expression during ALI culture is markedly delayed in cells derived from asthmatic donors compared with non-asthmatic donors. However, once HBE cells reach full differentiation (ALI days 14–21), no distinct differences are observed, as Hic-5 levels stabilize at a lower plateau following their reduction between ALI days 0–10. These findings are based on a limited dataset (two asthma donors) and require further validation in ongoing studies. Moreover, our in vitro observations align with re-analysis of published human scRNA-seq data from asthma exacerbation studies, which show increased Hic-5 expression 24 h after allergen exposure (**Fig. 1D**) but no baseline difference between allergic controls and allergic asthma. Together, these data suggest that elevated Hic-5 expression is not a constitutive feature of the asthmatic airway epithelium but instead reflects a transient, mechanically induced response to bronchoconstriction, which may serve as a regulatory mechanism in uncontrolled asthma. We have added this to the discussion to clarify this point (Lines 298- 305).

Comment 3. *Figure 2 demonstrates that mechanically-induced Hic-5 expression requires ERK and TGFb receptor signalling, and this expression is required for a robust transcriptional response to mechanical stress (Fig 3). But how does Hic-5 activate a transcriptional response? The study could benefit from immunofluorescence analysis of Hic-5 before and after compression. Several studies have demonstrated a translocation of Hic-5 at focal adhesions to the nucleus after stimuli, like H202 and mechanical stress (Knockdown of mechanosensitive adaptor Hic-5 ameliorates post-traumatic osteoarthritis in rats through repression of MMP-13 – doi.org/10.1038/s41598-023-34659-x ; Hic-5 Communicates between Focal Adhesions and the Nucleus through Oxidant-Sensitive Nuclear Export Signal – doi: 10.1091/mbc.02-06-*

0099). Is *Hic-5* actively activating transcription by entering the nucleus or does it signal through other mechanically responsive TFs, like YAP/TAZ or MRTFs?

Response 3:

We agree with the reviewer’s suggestion regarding immunofluorescence (IF) staining of *Hic-5*. To address this, we performed and now present IF staining in **Fig. 4A**. Under both control and compressed conditions, *Hic-5* localized predominantly in the cytosol and did not accumulate in the nucleus following mechanical compression, although its overall expression level increased.

For the reviewer’s convenience, we’ve included the newly added **Fig. 4A** below.

We also performed subcellular fractionation (**Fig. R2**). Although minor cytosolic contamination was present in the nuclear fraction, our data indicate that *Hic-5* did not accumulate in the nucleus but instead showed increased cytosolic abundance after compression, as demonstrated by IF staining. We have revised the results accordingly (Lines 230- 234).

Although Hic-5 has been reported to act as a co-activator of transcription factors such as the androgen receptor, our findings show that it does not accumulate in the nucleus of airway epithelial cells. This suggests that Hic-5 does not directly activate transcription via nuclear translocation during compression-induced mechanotransduction. Instead, it likely contributes indirectly by regulating cytoskeletal tension and actin dynamics, which can activate downstream mechanosensitive pathways such as YAP/TAZ or MRTF–SRF signaling^{2,3}. Both pathways are well established to respond to actin stress fiber assembly and cytoskeletal tension, raising the possibility that Hic-5 functions as an upstream regulator of these mechanotranscriptional programs in the airway epithelium. As our findings are the first to demonstrate the role of Hic-5 as a mechanoregulator in airway epithelial cells, further investigations are warranted to delineate the specific molecular pathways and transcriptional networks involved. To incorporate this insight, we accordingly revised the discussion (Lines 354 – 362).

Comment 4. The authors show in Fig 3 that Hic-5 is necessary for a robust transcriptional response to compression.

4A. Hic-5 is a homolog of Paxillin and found at focal adhesions, therefore, is this due to lack of Hic-5 or because focal adhesions have been disrupted and mechanotransduction cannot occur efficiently from the cell surface? Fig. 4 could benefit from immunostaining of some of the adhesion complexes that Hic-5 regulates.

4B. Additionally, the images in Fig. 4 are not quantified, which is something that they could do, based on their previous studies on jamming and unjamming.

4C. Also, is Hic-5 over-expression sufficient to drive a compression-like transcriptional response without compression?

Response 4:

We address each point (4A, 4B, and 4C) below:

4A. We agree that visualizing focal adhesion proteins such as FAK or vinculin would further strengthen our conclusion. This will be an important next step, which we plan to incorporate in future studies, particularly given the heterogeneity of primary cells and the need for protocol optimization across multiple donors. We appreciate this insightful suggestion by the reviewer.

4B. We appreciate the reviewer’s acknowledgment of our previous studies. In this work, we focused on identifying Hic-5 as a mechanoregulator, providing the first evidence of its role in human airway epithelial cells and its connection to asthma using data from previously published human studies. Our RNA-seq and ontology analyses further suggest that Hic-5 regulates stress fiber formation and ET-1 expression. Although we regret that the data collected were not sufficient to directly confirm its role in unjamming, time-lapse imaging for all experimental conditions could not be completed during the pandemic. In response to the reviewer’s suggestion, using five donors described in Fig. 4B, we quantified the cell shape index as present in Extended Data Figure 8. This analysis suggests that Hic-5 may contribute to the unjamming transition (UJT) in response to mechanical compression, although additional studies combining UJT assessment with cell migration analyses will be required to validate this finding.

4C. We appreciate this insightful question, which is important for clarifying the role of Hic-5 as a transcriptional regulator. We did not explicitly test whether Hic-5 overexpression (OE) alone is sufficient to induce a compression-like transcriptional program in the absence of compression. Based on our findings and prior work, we speculate that Hic-5 OE alone is unlikely to be sufficient. We have two reasons. First, as shown in **Fig. 4A** and discussed in Response 3 above, Hic-5 did not accumulate in the nucleus, suggesting that its transcriptional effects likely result from Hic-5-mediated cytoskeletal regulation rather than direct nuclear action. Second, while Hic-5 is necessary for stress fiber formation, OE may not be sufficient, as supported by previous studies. For example, in vimentin-rich fibroblasts, Hic-5 acts primarily as a focal adhesion adaptor, facilitating tension-dependent cytoskeletal remodeling via vimentin rather than initiating it de novo⁴. Together, these findings support the view that Hic-5 mediates mechanotransduction but is not sufficient on its own. We have noted this point in the Discussion (Lines 354–367).

Comment 5.

5A. *In the title the authors say that Hic-5 drives a vicious cycle of bronchoconstriction, one way being the stimulated secretion of ET-1 (Fig 5) in response to compression. However, the authors do not show any direct data relating to bronchoconstriction and it is first referenced in line 235-237 (Ref 28). Would the authors be able to either highlight or expand on this data to further support their title? If not, we recommend editing the title to not overstate the findings.*

5B. *Also, in Fig 5 Hic-5 KD shows near ablation of EDN1 mRNA after compression, however there is still a significant elevation of ET-1 protein after compression when comparing control and Hic-5 KD. Can the authors propose why there might still be residual ET-1 detected in Hic-5 KD after compression?*

5C. *Finally, the finding that ET-1 is much more tightly regulated by Hic-5 under compression without any smooth muscle is a finding that needs further highlighting in the discussion, as it could be the basis for therapeutic intervention.*

Response 5:

We address each point (5A, 5B, and 5C) below:

5A. We understand the reviewer's concern. To avoid overstating our findings, we have revised the title to: Hic-5 drives epithelial mechanotransduction, promoting a feed-forward cycle of bronchoconstriction.

5B. We thank the reviewer for this careful observation. In Fig. 5B, compression-induced EDN1 mRNA is significantly reduced, but not fully ablated, by Hic-5 KD. This is consistent with the residual Hic-5 protein that remained detectable, although markedly reduced as shown in **Fig. 3A**. As our approach was knockdown (KD) rather than a complete knockout (KO), the residual Hic-5 likely led to a modest ET-1 induction, reflected by the small increases in *EDN1* mRNA (**Fig. 5B**) and secreted ET-1 (**Fig. 5C**) after compression. From a translational perspective, this partial—but substantial—suppression may be advantageous for an ASO-based approach, as it could maintain basal expression needed for homeostasis while attenuating the excessive induction seen under mechanical stress.

For clarification, we have included raw qPCR data for *EDN1* mRNA expression below. In addition, to avoid confusion, we have revised the mRNA expression graphs (both **Fig 5A and 5B**) by presenting fold change for compression-induced *EDN1* normalized to the corresponding control condition in Hic-5 WT and KD cells.

		EDN1					GAPDH					
		Ct-1	Ct-2	Ct-3	Ct-avg	St. Dev.	Ct-1	Ct-2	Ct-3	Ct-avg	St. Dev.	
U19	Hic-5 WT	C1	25.28	25.29	25.37	25.31	0.05	18.54	18.51	18.58	18.54	0.04
		C2		25.11	25.22	25.16	0.08	18.38	18.96	18.84	19.06	0.29
		P1	23.59	23.60	23.68	23.62	0.05	19.28	19.03	18.96	19.09	0.17
	Hic-5 KD	P2	23.20	23.15	23.20	23.18	0.03	18.95	18.76	18.62	18.78	0.17
		C1	30.45	30.76	30.68	30.63	0.16	23.72	23.70	23.79	23.73	0.05
		C2	26.94	26.88	27.04	26.95	0.08	20.85	20.76	20.68	20.77	0.08
U16	Hic-5 WT	P1	29.93	30.08	30.23	30.08	0.15	23.32	23.34	23.33	23.33	0.01
		P2	30.58	30.41	31.01	30.67	0.31	23.83	24.24	24.10	24.06	0.21
		C1	27.23	27.05	26.95	27.07	0.14	19.25	19.15	19.33	19.24	0.09
	Hic-5 KD	C2	27.02	26.96	27.17	27.05	0.11	18.86	18.95	18.89	18.90	0.04
		P1	23.72	23.72	23.86	23.77	0.08	18.67	18.59	18.65	18.64	0.04
		P2	24.00	23.88	23.88	23.92	0.07	18.85	18.87	18.89	18.87	0.02
U30	Hic-5 WT	C1	25.60	25.63	25.70	25.64	0.05	19.53	19.44	19.49	19.49	0.04
		C2	25.85	25.74	25.91	25.84	0.08	19.48	19.41	19.46	19.45	0.04
		P1	26.04	26.09	26.11	26.08	0.03	19.39	19.45	19.48	19.44	0.04
	Hic-5 KD	P2	26.47	26.37	26.55	26.53	0.05	20.20	20.12	20.20	20.17	0.04
		C1	28.48	28.50	28.54	28.51	0.03	19.10	19.07	19.16	19.11	0.04
		C2	28.30	28.27	28.49	28.35	0.12	19.26	19.31	19.31	19.29	0.03
Hic-5 WT	P1	24.25	24.34	24.38	24.33	0.06	19.16	19.21	19.39	19.25	0.12	
	P2	24.27	24.27	24.32	24.28	0.03	19.47	19.21	19.17	19.28	0.17	
	C1	30.83	30.60	30.98	30.80	0.19	23.24	23.25	23.35	23.28	0.06	
Hic-5 KD	C2	29.03	29.30	29.21	29.18	0.14	21.90	21.90	21.55	21.66	0.03	
	P1	27.43	27.60	27.68	27.57	0.13	20.99	21.05	21.04	21.03	0.03	
	P2	27.49	27.30	27.61	27.47	0.16	20.75	20.72	20.84	20.77	0.06	

		EDN1						
		dCT	dCT-control	ddCT	RQ (2 ^{-ddct})	Fold change	St. Dev	
U19	Hic-5 WT	C1	6.77	6.44	0.33	0.79	1.03	0.33
		C2	6.10	6.44	-0.33	1.26	3.92	0.23
		P1	4.53	6.44	-1.91	3.75	4.08	
	Hic-5 KD	P2	4.41	6.44	-2.03	4.08	1.03	0.35
		C1	6.89	6.54	0.36	0.79	1.03	0.35
		C2	6.18	6.54	-0.36	1.28	0.91	0.06
U16	Hic-5 WT	P1	6.75	6.54	0.21	0.87	1.01	0.16
		P2	6.81	6.54	0.27	0.95		
		C1	7.83	7.99	-0.16	1.12		
	Hic-5 KD	C2	8.15	7.99	0.16	0.89		
		P1	5.13	7.99	-2.86	7.25	7.47	0.31
		P2	5.05	7.99	-2.94	7.69		
U30	Hic-5 WT	C1	6.16	6.27	-0.12	1.08	1.00	0.11
		C2	6.39	6.27	0.12	0.92		
		P1	6.64	6.27	0.37	0.77	0.86	0.12
	Hic-5 KD	P2	6.35	6.27	0.08	0.94		
		C1	9.40	9.23	0.17	0.89	1.01	0.17
		C2	9.06	9.23	-0.17	1.12		
Hic-5 WT	P1	5.07	9.23	-4.16	17.86	18.31	0.63	
	P2	5.00	9.23	-4.23	18.75			
	C1	7.52	7.57	-0.05	1.03	1.00	0.05	
Hic-5 KD	C2	7.62	7.57	0.05	0.97			
	P1	6.54	7.57	-1.03	2.04	1.93	0.15	
	P2	6.70	7.57	-0.87	1.83			

5C. We can't agree more with the reviewer and appreciate the reviewer's insights. To highlight this point, we have added text to the Discussion (Lines 345 – 352), as follows:

In the pseudostratified human airway epithelium, mechanical compression mimicking bronchoconstriction increases ET-1 secretion independently of airway smooth muscle, suggesting that epithelial mechanotransduction alone is sufficient to drive ET-1 secretion. Hic-5 KD markedly suppresses this induction, supporting a role for Hic-5 as a critical regulator of epithelial ET-1 secretion under mechanical stress. These findings raise the possibility that a Hic-5–ET-1 axis contributes to a feed-forward cycle of bronchoconstriction in severe asthma and provide a rationale to explore epithelial-specific Hic-5 inhibition, for example through inhaled ASO therapy.

Reviewer #2 (Remarks to the Author):

Comment: *I co-reviewed this manuscript with one of the reviewers who provided the listed reports. This is part of the Nature Communications initiative to facilitate training in peer review and to provide appropriate recognition for Early Career Researchers who co-review manuscripts.*

Response: We appreciate the reviewer's time and effort in reviewing our manuscript.

Reviewer #3 (Remarks to the Author):

Comment: *I co-reviewed this manuscript with one of the reviewers who provided the listed reports. This is part of the Nature Communications initiative to facilitate training in peer review and to provide appropriate recognition for Early Career Researchers who co-review manuscripts.*

Response: We appreciate the reviewer's time and effort in reviewing our manuscript.

Reviewer #4 (Remarks to the Author):

General comment:

...

The authors conclude that *Hic-5* is a key regulator of mechanotransduction in airway epithelial cells and this links bronchoconstriction-induced mechanical stress to asthma pathogenesis. The data are clearly presented, and the manuscript is well written. The overall demonstration that *Hic-5* is responsive to airway epithelial cell compression *in vitro* is convincing. The data from human studies does provide evidence that *Hic-5* is expressed in basal cells and upregulated following an asthma-relevant stimulus, but it cannot be concluded that this is due to bronchoconstriction. The *Hic-5* knockdown is impressive in the scope of effects, but the details of the methodology are limited. It is also not clear how well compression of cultures mimics the situation with bronchoconstriction *in vivo*. Finally, there is very little mechanistic data on how *Hic-5* upregulates endothelin-1/PDGF β expression. Overall, the authors have identified an interesting mechanism driving *Hic-5* expression that may link bronchoconstriction to airway remodeling, but the data are not fully supportive of their conclusions.

Response:

We are grateful to the reviewer for taking the time and effort to review our manuscript. We especially appreciate the insightful comments provided in the general section, which we have thoroughly addressed in the specific comments section. Below, we specified three points raised in reviewer's general comments and link them to the corresponding response numbers addressed in the specific comments section.

1. "The data from human studies does provide evidence that *Hic-5* is expressed in basal cells and upregulated following an asthma-relevant stimulus, but it cannot be concluded that this is due to bronchoconstriction": Response 2.
2. "The *Hic-5* knockdown is impressive in the scope of effects, but the details of the methodology are limited.": Response 4.
3. "... there is very little mechanistic data on how *Hic-5* upregulates endothelin-1/PDGF β expression": Response 7

Specific comments

Comment 1. For the human correlation, the authors state that TGF β 111 levels were increased post-allergen in asthmatics but not in the control subjects. Are these data available? Were the differences significant?

Response 1:

Thank you for raising this important point. While we described these findings in the Results of the submitted manuscript, we realized that the corresponding data were inadvertently omitted from the original submission. In the revised version, we have now included these data as **Extended Data Figure 2**.

Comment 2. Allergen challenge obviously induces multiple processes, including bronchoconstriction. How can you definitely say it was the bronchoconstriction in human subjects that induced the TGF β 111?

Response 2:

This comment is also related to the reviewer’s general comment noting that “**The data from human studies provide evidence that *Hic-5* is expressed in basal cells and upregulated following an asthma-relevant stimulus, but it cannot be concluded that this is due to bronchoconstriction.**”

We appreciate the reviewer’s concern and agree that allergen challenge in vivo induces multiple processes. Our interpretation that bronchoconstriction contributes to *TGFB111* induction is based on both our *in vitro* studies and published human data. In our *in vitro* system, *TGFB111* was induced by mechanical compression alone, which mimics asthmatic bronchoconstriction. We have previously shown that this model recapitulates key hallmarks of airway remodeling observed in patients with asthma, and these findings have been further validated by human studies demonstrating that bronchoconstriction can drive airway remodeling⁵.

Consistent with this, analysis of published single-cell RNA-seq data from allergen-challenge studies revealed that *TGFB111* was upregulated predominantly in basal epithelial cells of individuals with asthma who developed bronchoconstriction (allergic asthma, AA), but not in allergic individuals without asthma (allergic controls, AC) (Fig. R4). Importantly, no baseline differences were detected between allergic controls and allergic asthma subjects, indicating that elevated *TGFB111* may not be a constitutive feature of the asthmatic airway epithelium, but transiently increased under specific conditions, including bronchoconstriction.

While we cannot exclude contributions from other processes triggered by allergen challenge, the concordance between our *in vitro* compression model and the human scRNA-seq data strongly supports bronchoconstriction as a key driver of *TGFB111* induction during asthma exacerbations. To highlight this and improve the clarity of our findings, we’ve revised the discussion (Lines 298-304).

Comment 3. More details on the method of cellular compression (force used, % compressed etc) would be helpful. How well does this mimic the forces in the airway of an asthmatic that is constricted?

Response 3:

The details of cellular compression are described in the Methods section under the subheading “*In vitro* exposure of HBE cells to mechanical compression, TGF-β1, or inhibitors.” In the Methods, we provide both the magnitude of cellular compression and duration of exposure.

We apply 30 cm H₂O pressure, which presents compressive mechanical stress comparable to the stress exerted by constricted airway smooth muscle during maximal bronchoconstriction^{6,7}. During maximal bronchoconstriction, airway epithelial cells are subjected to compressive stress of approximately 30 cm H₂O, which is at least an order of magnitude greater than the transepithelial stress experienced during normal breathing. This method of cellular compression has been established as presented in our previous work, including numerous original research articles⁸⁻¹⁵ and review articles^{16,17}. Moreover, our *in vitro*

findings using cellular compression were validated by human studies demonstrating that bronchoconstriction induces airway remodeling⁵.

Comment 4. *What are the antisense oligonucleotides utilized? How were they introduced into the cells and how specific are they to TGFB111? Could there be off-target effects?*

Response 4:

This comment is also related to the reviewer's general comment noting that "**The Hic-5 knockdown is impressive in the scope of effects, but the details of the methodology are limited.**"

We thank the reviewer for acknowledging the impressive Hic-5 KD efficiency, and we agree that providing more details would benefit other researchers. In response to this comment regarding the lack of methodological details, we have revised the Methods section to include additional information, as shown below. We also added a critical reference (PMID: 35487895)¹⁸ describing the use of ASOs in HBE cells by colleagues of our co-author at Ionis Pharmaceuticals.

Hic-5 knockdown using antisense oligonucleotides (ASOs)

To knock down Hic-5 expression in well-differentiated primary HBE cells, we used antisense oligonucleotides (ASOs) designed and provided by Ionis Pharmaceuticals, following the previously established ASO approach¹⁸. We initially screened five ASOs targeting the coding region of human TGFB111 (denoted as Hic-5 ASO). We then selected the most effective ASO based on the reduction of Hic-5 expression determined by both RT-qPCR and western blot analysis (**Extended data Fig. 3**). We used this selected ASO targeting Hic-5 for subsequent experiments. As a non-targeting control, we used a scrambled ASO (denoted as control ASO) provided by Ionis. We reconstituted lyophilized ASOs in PBS to 5 mM, aliquoted the stock ASO, and stored the aliquots at -80 °C until use. During ALI culture, we replenished basal medium containing 10 uM ASO (scrambled or Hic-5) every other day from ALI day 9 to day 19.

To address the concern regarding potential off-target effects, we have now included a volcano plot (**Extended Data Figure 6**) demonstrating differentially expressed genes (DEGs) between Hic-5 WT and KD HBE cells under control conditions. RNA-seq analysis revealed no other DEGs between Hic-5 WT and KD cells, except for the significantly downregulated *TGFB111*, which encodes the Hic-5 protein (**Extended Data Fig. 6**). This result confirms the selectivity of the ASO-mediated Hic-5 knockdown with minimal off-target effects. We have revised the results section to include this supplementary information (Lines 183-187).

Extended Data Figure 6. The Volcano plot presents differentially expressed genes (DEGs) between Hic-5 WT and KD HBE cells under control conditions. Only Hic-5 was significantly reduced in Hic-5 KD HBE cells.

Comment 5. *It would be helpful to quantify the band intensity in the western blots shown in Figures 2 and 3.*

Response 5:

We thank the reviewer for this suggestion. We have quantified the band intensities of the western blots in **Figs 2 and 3** using densitometry analysis, and both figures have been updated accordingly.

The updated figures with densitometry analysis includes **Fig. 2C, 2D, 2E** and **Fig. 3A**.

Comment 6. Is Figure 1A necessary, given the more quantitative data in 1B? Why not show the time course?

Response 6:

Fig. 1A is necessary because it demonstrates the RNA-seq data using the cells from six donors, while **Fig. 1B** provides validation of these results using qPCR in additional three donors. **Fig. 1B** is from one time point, but it was based on a time course (3, 8, and 24 hours) experiment, which showed a similar trend to the RNA-seq data. Based on the time-point experiment from both RNA-seq and qPCR analysis, we selected the 3-hour time point for subsequent mRNA detection. To address the reviewer's concern and provide additional information to the reader, we have now included the time course data as **Extended Data Fig. 1**.

Comment 7. Are *EDN1* and *PDGFbeta* upregulated in the human data, similar to *TGFB11*?

Response 7:

This comment is also related to the reviewer's general comment noting that "... there is very little mechanistic data on how *Hic-5* upregulates endothelin-1/*PDGFbeta* expression."

We understand that the reviewer's concern regarding the limited mechanistic studies. These are critical research questions, but they are beyond the scope of the current work. We hope to address this critical question in follow-up studies that build on this current studies.

To address the reviewer's specific question on the upregulation of *EDN1* and *PDGFB*, here we share re-analyzed data with the reviewer (**Fig. R5**). *EDN1* expressed in basal cells was significantly induced following an allergen challenge, which is similar to the regulation of *TGFB11*. However, *PDGFB* expression was not comparable to that of *TGFB11*. These human data further suggest shared signaling between *Hic-5* and *EDN1* expression or a potential *Hic-5*-mediated regulation of *EDN1* in human airway epithelial cells.

Figure R5. Re-analysis of published scRNA-seq data from human bronchial biopsies¹⁶. Each plot presents the percentage of cells (y-axis) of each epithelial cell type (x-axis) expressing annotated gene in individual patients (small dots) after challenge with an allergenic antigen (red) or diluent control (gray).

Comment 8. Can you link any of the known activities of Hic-5 to the signaling changes noted?

Response 8:

One of the best-characterized roles of Hic-5 is in promoting stress fiber formation. Our data support this role, as we observed actin stress fiber formation in a Hic-5-dependent manner under mechanical compression. Previous studies have shown that Hic-5 mediates stress fiber formation through a RhoA/ROCK-dependent pathway¹⁹, and we suspect that a similar mechanism may be implicated in our system.

Comment 9. It is unclear to me why compressed WT cells don't have more dramatic DEG compared to compressed Hic-5 KO cells (Fig 3D), given the dramatic differences in the profiles when compared to control cells.

Response 9:

We appreciate the reviewer's comment and recognize that **Fig. 3D** may not have been sufficiently clear. The number of differentially expressed genes (DEGs) between control and compressed cells is much greater in WT (**Fig. 3B**) than in KD (**Fig. 3C**). By contrast, **Fig. 3D** compares compressed WT with compressed KD cells. These DEGs were then used for downstream analyses, including gene ontology and pathway enrichment, as presented in **Fig. 3E** and **3F**. To avoid confusion, we have clarified this point in the revised manuscript (Lines 196–201).

References

- 1 Kobrossy, L. *et al.* Unraveling MLL1-fusion leukemia: Epigenetic revelations from an iPS cell point mutation. *J Biol Chem* **300**, 107825 (2024). <https://doi.org/10.1016/j.jbc.2024.107825>
- 2 Dupont, S. *et al.* Role of YAP/TAZ in mechanotransduction. *Nature* **474**, 179-183 (2011). <https://doi.org/10.1038/nature10137>
- 3 Finch-Edmondson, M. & Sudol, M. Framework to function: mechanosensitive regulators of gene transcription. *Cell Mol Biol Lett* **21**, 28 (2016). <https://doi.org/10.1186/s11658-016-0028-7>
- 4 Vohnoutka, R. B. *et al.* The focal adhesion scaffold protein Hic-5 regulates vimentin organization in fibroblasts. *Mol Biol Cell* **30**, 3037-3056 (2019). <https://doi.org/10.1091/mbc.E19-08-0442>
- 5 Grainge, C. L. *et al.* Effect of bronchoconstriction on airway remodeling in asthma. *N Engl J Med* **364**, 2006-2015 (2011). <https://doi.org/10.1056/NEJMoa1014350>
- 6 Wiggs, B. R., Hrousis, C. A., Drazen, J. M. & Kamm, R. D. On the mechanism of mucosal folding in normal and asthmatic airways. *J Appl Physiol (1985)* **83**, 1814-1821 (1997).
- 7 Yager, D. *et al.* Amplification of airway constriction due to liquid filling of airway interstices. *J Appl Physiol (1985)* **66**, 2873-2884 (1989).
- 8 Mwase, C. *et al.* TGF-beta Receptor-dependent Tissue Factor Release and Proteomic Profiling of Extracellular Vesicles from Mechanically Compressed Human Bronchial Epithelial Cells. *Am J Respir Cell Mol Biol* (2025). <https://doi.org/10.1165/rcmb.2024-0130OC>
- 9 Mitchel, J. A. *et al.* In primary airway epithelial cells, the unjamming transition is distinct from the epithelial-to-mesenchymal transition. *Nat Commun* **11**, 5053 (2020). <https://doi.org/10.1038/s41467-020-18841-7>
- 10 Kilic, A. *et al.* Mechanical forces induce an asthma gene signature in healthy airway epithelial cells. *Sci Rep* **10**, 966 (2020). <https://doi.org/10.1038/s41598-020-57755-8>
- 11 Kim, S. H. *et al.* Elevated extracellular maspin after mechanical compression in vitro or allergen challenges in vivo. *J Allergy Clin Immunol* (2019). <https://doi.org/10.1016/j.jaci.2019.06.006>

- 12 Lan, B. *et al.* Airway epithelial compression promotes airway smooth muscle proliferation and contraction. *Am J Physiol Lung Cell Mol Physiol* **315**, L645-L652 (2018).
<https://doi.org/10.1152/ajplung.00261.2018>
- 13 Mitchel, J. A. *et al.* IL-13 Augments Compressive Stress-Induced Tissue Factor Expression in Human Airway Epithelial Cells. *Am J Respir Cell Mol Biol* **54**, 524-531 (2016).
<https://doi.org/10.1165/rcmb.2015-0252OC>
- 14 Park, J. A. *et al.* Unjamming and cell shape in the asthmatic airway epithelium. *Nat Mater* **14**, 1040-1048 (2015). <https://doi.org/10.1038/nmat4357>
- 15 Park, J. A. & Tschumperlin, D. J. Chronic intermittent mechanical stress increases MUC5AC protein expression. *Am J Respir Cell Mol Biol* **41**, 459-466 (2009).
<https://doi.org/10.1165/rcmb.2008-0195OC>
- 16 O'Sullivan, M. J., Phung, T. N. & Park, J. A. Bronchoconstriction: a potential missing link in airway remodelling. *Open Biol* **10**, 200254 (2020). <https://doi.org/10.1098/rsob.200254>
- 17 Park, J. A., Fredberg, J. J. & Drazen, J. M. Putting the Squeeze on Airway Epithelia. *Physiology (Bethesda)* **30**, 293-303 (2015). <https://doi.org/10.1152/physiol.00004.2015>
- 18 Sanderlin, E. J. *et al.* CFTR mRNAs with nonsense codons are degraded by the SMG6-mediated endonucleolytic decay pathway. *Nat Commun* **13**, 2344 (2022).
<https://doi.org/10.1038/s41467-022-29935-9>
- 19 Tumbarello, D. A. & Turner, C. E. Hic-5 contributes to epithelial-mesenchymal transformation through a RhoA/ROCK-dependent pathway. *J Cell Physiol* **211**, 736-747 (2007).
<https://doi.org/10.1002/jcp.20991>

Response to the reviewer

Reviewer #4

Comment: The revised manuscript and response to review have addressed almost all of my comments. One issue needs to be further clarified:

In the new extended data Figure 2 it appears that the AC subjects also have significant upregulation in TGFB111 in the allergen sample (p value listed is 0.03). The values look higher in the AA but its unclear if there is a difference between AA and AC. The legend implies that there is no change in AC which is not correct based on a p value of 0.03. This needs to be discussed in more detail. Similarly, I don't think the authors are acknowledging that there may be other factors with an allergen challenge that effect Hic-5. I think the discussion needs to acknowledge that the human allergen challenge data is merely suggestive and other factors induced by allergen may increase Hic-5, especially in light of the data now shown in the Extended figure 2.

Response:

We appreciate the reviewer's thorough review of our data. We oversighted and agreed with the reviewer.

To reflect the data presented in Supplementary Fig. 2, we have made the following three changes.

- revised Supplementary Fig. 2 to address reviewer's question: AA vs AC ($p= 0.0046$).
- revised the text in the result section (lines 107-109).
- revised the discussion to acknowledge other factors with an allergen challenge (lines 298- 301).